# Reasons for (not) choosing dental treatments—A qualitative study based on patients' perspective

**Susanne Felgner●\*, Marie Dreger, Cornelia Henschke●**

Department of Health Care Management, Berlin Centre of Health Economics Research Technische Universität Berlin, Berlin, Germany

\* susanne.felgner@tu-berlin.de

## Abstract

Oral health is increasingly seen as a public health challenge due to the remarkable prevalence of oral diseases worldwide, the impact on general health, and health consequences that can arise for individuals. Compared to other health services, oral health services are usually not fully covered by statutory health insurance, which is seen as one reason in decision-making on dental treatments. Nevertheless, patients' reasons for treatment decisions are not well understood although they can provide valuable insights. The objective of this study was to identify reasons of choice for dental treatments and to explore patients' view on cost coverage in Germany. We conducted four focus group interviews with a total of 27 participants. The interviews were audiotaped and transcribed verbatim. Data was analyzed performing conventional content analysis. As part of a qualitative analysis, subcategories and categories were formed from identified reasons using an inductive approach. Our study supports and expands research in exploring patients' decision-making on dental treatments. It highlights a variety of 53 reasons of choice for dental treatments from patients' perspective, split in two categories "health care service", and "dentist & dental office". First category includes reasons regarding dental care performance (subcategories: "preconditions", "treatment", "costs", and "outcomes"). Second category demonstrates reasons regarding dentists, office structures and processes (subcategories: "professional skills", "social skills", "office staff & equipment", and "office processes"). Reasons named "most important" by the participants are out-of-pocket payments, dentists' training, and a relationship of trust between patient and dentist. Although the participants use incentive measures to lower financial burden, several perceived challenges exist. Identified reasons for choosing dental treatments provide a basis for further studies to quantify the relevance of these reasons from patients' perspective. Based on this, the various reasons identified can be considered in future policies to improve patients' utilization behavior, which can range from improved information sources to increased incentive measures.

**Data Availability Statement:** All relevant data that can be publicly displayed are available within the paper and its Supporting information files. We do not offer access to anonymized interview transcripts in another format, extend, or on

demand. Non-aggregated participants information and transcripts may not be publicly shared due to ethical restrictions, approved by the ethical committee of the Technische Universität Berlin in Berlin, Germany.

**Funding:** This study was funded through the Berlin Centre for Health Economics Research by the German Federal Ministry of Education and Research (grant no. 01EH1604A), URL: www. bmbf.de. All authors (SF, MD, CH) are grant recipients. The funder had no role in study design, data collection and analysis, decision to publish, or preparation of the manuscript.

**Competing interests:** The authors have declared that no competing interests exist.

## Introduction

Oral health, including dental health, has become a global public health challenge. It is known to affect the overall health of individuals causing further health complaints [1], but also social and psychological concerns [2, 3]. The worldwide prevalence of chronic and progressive dental diseases is remarkable, e.g., an estimated 2.3 billion people suffer from caries of permanent teeth [4]. Chronic untreated dental diseases can be associated with serious consequences including pain, sepsis, reduced quality of life, and decreased work productivity. The burden for patients and the healthcare system is increasing also from an economic point of view [5]. For patients, treatments of dental diseases and tooth loss are accompanied by high co-payments (out-of-pocket payments) as costs are often not (fully) covered by statutory health insurance (SHI), especially in the case of dentures [6, 7]. Studies have shown that coverage of dental treatments by health insurances can impact patients' utilization of dental services [8].

Although there is broader coverage of dental care treatments in Germany compared to other countries [9, 10], oral health outcomes are not superior in countries such as Spain and the Netherlands [11]. Therefore, identifying factors that determine decision-making on dental treatments can help to understand patients' perspective and their decisions regarding their choice of dental treatments. It can provide a basis for further analysis that may reveal a need for changes in policies and practice.

Previous studies considered costs, long duration times and patients' fear as reasons of choice for dental treatments in different countries, e.g., Saudi-Arabia [12], and the USA [13]. Some existing studies are limited to certain dental services, e.g., preventive measures [14], and caries treatment [15]. However, a basic understanding of possible reasons of choice for dental treatments from patients' perspective is limited, especially in Germany. Although this study focuses on the German health care setting, it contributes to the identification of reasons that may influence a decision for dental treatments.

To explore and better understand patients' choice for dental treatments these considerations led us to the following research questions:

1. *What reasons do patients have for choosing or not choosing dental treatments?*

2. *How do patients assess cost coverage of dental services by statutory health insurance?*

## Coverage of dental services and in particular dentures in Germany

In the following, coverage of dental care and in particular dentures by SHI are described to give a better understanding of the regulatory framework in German dental care.

Around 87% of the German citizens are insured by the SHI [16]. Although many basic treatments such as dental check-ups and tooth-fillings [17] are covered fully by SHI, treatments exceeding standard services as well as dentures are only covered in parts [18]. Exceptions include a range of services for patients aged under 18 and certain cases of orthodontic treatments [19]. Usually, costs exceeding standard care must be paid out-of-pocket by patients [20]. Coverage for dentures is more restricted compared to other services in health care using fixed subsidies that differ between predefined groups (standard, similar, and different treatments) (Fig 1). Similar treatments differ from standard care, for instance in terms of dental materials used for a crown. A different treatment includes a different type of restoration compared to standard care. According to these, patients' out-of-pocket payments vary.

Standard care for dentures (medically necessary treatment) is described in the Uniform Value Scale for Dental Services (*short in German*: BEMA) [21] and the Federal Uniform List for Dental Technical Performances (*short in German*: BEL II– 2004) including dentist's and dental technician's performances for certain treatments [20]. A subsidy of 60% of the

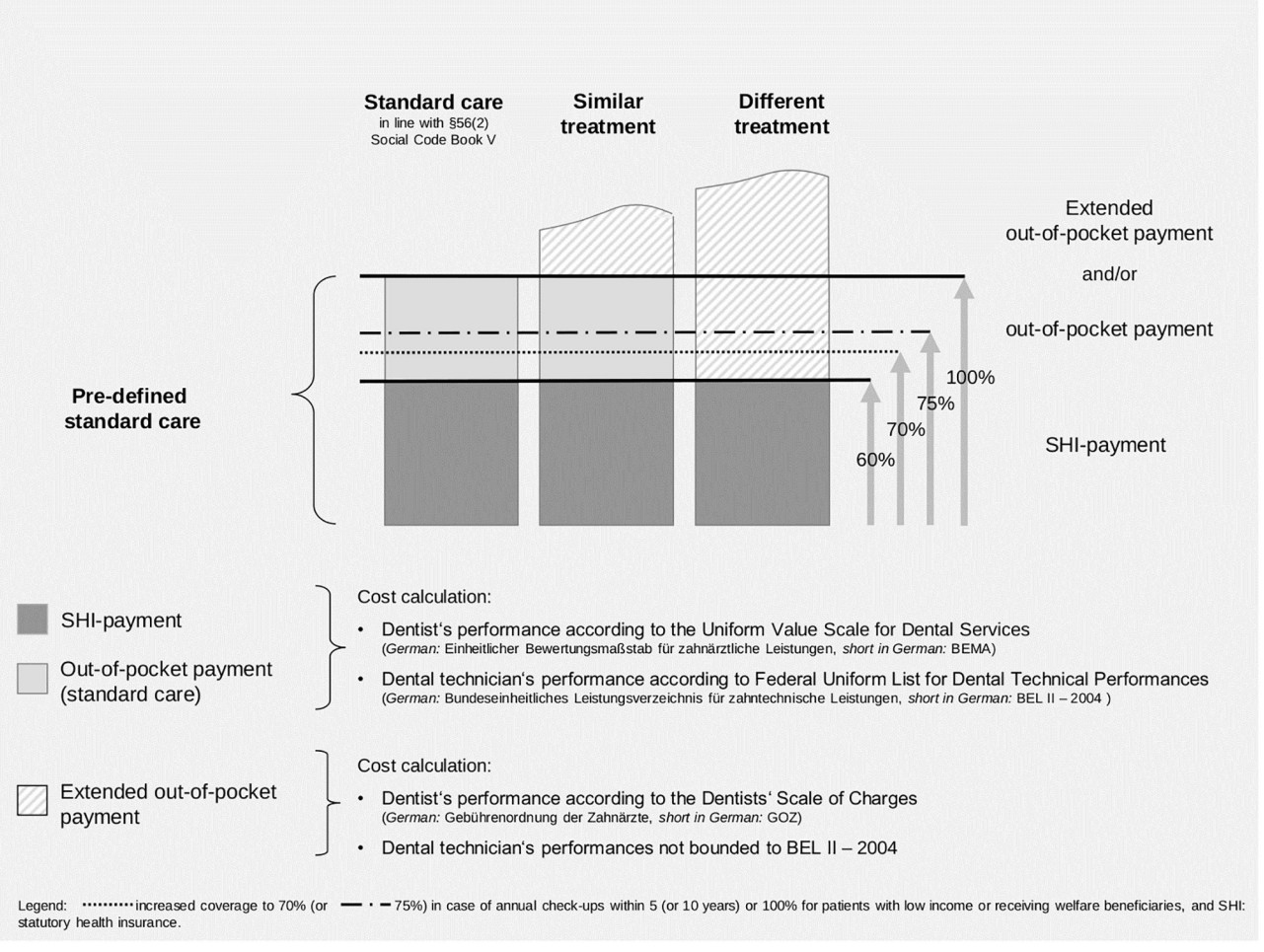

**Fig 1. Coverage of dentures in the German statutory health insurance.**

treatment costs of pre-defined standard care is covered by SHI (until September 2020: 50%). If patients decide for a standard care treatment, they must pay 40% of defined standard care costs (before October 2020: 50%) out-of-pocket. If patients decide for a similar treatment or a different treatment, they also must pay 40% (before October 2020: 50%) of the standard care costs plus the difference of costs between their chosen treatment alternative and predefined standard care (extended out-of-pocket payment) [22]. For the latter one, dentists set their prices according to a price list for privately delivered dental services (Catalogue of Tariffs for Dentists, *short in German*: GOZ) [23].

Co-payments decrease if patients use regular preventive check-ups documented in a "Bonus booklet". Annual dental check-ups within five years (or ten years) prior to the respective treatment increase SHI's subsidy to 70% (or 75%) of the cost of the predefined standard care [before October 2020: 60% (or 65%)] [20, 22]. Additionally, so called "Bonus programs" allow individual health insurances of SHI to offer further incentives to its members [24]. Patients may also use supplementary insurances covering costs not considered by SHI subsidies. Regarding financial protection measures, there is an extended coverage up to 100% of standard care for welfare beneficiaries or patients with low income [20, 25].

Beside SHI, around 11% of the population is covered through private health insurance (PHI). PHI is an optional alternative for self-employed and employees above a certain income threshold. Coverage of dental care in SHI depends heavily on individual insurance contracts. Private insurances and its patients therefore pay different prices for dental treatments compared to SHI [14].

## Materials and methods

### Study design and recruitment of participants

Designed as a qualitative study, we conducted focus group interviews planned with a maximum of ten participants each, as recommended in scientific literature [26]. Initially, four interviews were scheduled with the option to conduct more if saturation point was not reached within analysis [26]. We followed the Consolidated Criteria for Reporting Qualitative Research (COREQ) to ensure reporting quality of our focus group interviews (S1 File) [27]. Using different communication channels, participants were recruited through online media [Facebook, eBay, mailing list of the Technische Universität Berlin (TUB)], and print media (in all districts of Berlin: flyers in grocery stores, free local weekly newspaper) to reach a diverse group of respondents (i.e., all ages, educational and income statuses).

For participation, a compensation of 20€ was announced. Following the approach of purposive sampling, we included interested people who met the following criteria: minimum age of 18 years, and a SHI or PHI membership. Appointment dates could be chosen by the potential participants according to predefined selection dates. Our study was approved by the ethical committee of the TUB (Ref. FEL_01_20170320).

### Procedure

Before conducting the interviews, one pilot interview was performed allowing us to refine the developed guide. The methodical structure of the interviews was chronologically and consistently arranged including (a) opening and introductory questions, (b) transition questions, (c) key questions, and (d) ending questions [26]. All interviews were moderated and co-moderated by the same female researchers [SF (MSc), CH (PhD)] in the same facilities at TUB to ensure consistency in the processes. Both were experienced in conducting qualitative studies [28, 29]. A student assistant (MD, female; FR, male) was responsible for recording the order of the speakers to allow for allocating statements to the participants. No further persons were present during the interviews. All interviews were audiotaped. The research objectives were explained to the (potential) participants during the recruitment process by means of online information and flyers, and orally at the beginning of the interviews (plus an information document). Since researchers and the student assistant were unknown to the participants, they briefly introduced themselves mentioning their employment at TUB, education, and main areas of research. SF and CH have been research fellows at the Department of Health Care Management. During the conduction of interviews MD was employed at the department as student assistant. Later on, she was part of the team as a research fellow (MSc). No interview guide or questions were provided before the interviews, and no transcripts were returned afterwards to the participants.

The interviews consisted of two parts:

1. The participants were asked to report their experiences at dental offices and explain reasons for their decisions for or against dental treatments. During a 20-minute break, the discussion protocol of each focus group interview was screened by the two researchers to

summarize statements belonging to the same content to identify reasons for (not) choosing dental treatments that were inductively derived from it.

2. Subsequently, findings were presented verbally and visually on a chart board to the participants with the request to confirm, reject and complete defined reasons representing their expectations in dental care. Afterwards, each participant was asked to name her/his three most important reasons when choosing a dental treatment. The number of mentions was counted for each reason. Finally, the participants anonymously answered a questionnaire on their socio-demographic characteristics, including information on their (supplementary) insurance, and signed the informed consent. Then, the compensation was paid out. For ensuring overall anonymization, the participants were orally addressed by their first or a self-chosen name during the focus groups interviews. The participants' information document, the questionnaire, the informed consent, and the interview guide can be found in the S2–S4 Files, and S1 Table.

## Data editing and analysis

We performed three steps of data analysis built on each other: (1) identification and coding of reasons of choice regarding dental treatments, and summarizing these in evolved (sub)categories, (2) assignment of reasons to the subject of choosing and not choosing dental treatments, and (3) determination of the thoughts of cost coverage from patients' perspective. An external office transcribed the interviews literally. Names and further information allowing conclusions about the participants were replaced by numbers and common terms.

The three analysis steps are presented in more detail in the following:

1. We performed conventional content analysis based on a developed coding scheme and codebook using the software Atlas.ti (version 7). Through this inductive approach of content analysis, codes of reasons of choice and (sub)categories emerged from the data, and findings moved from specific to general [30, 31]. Content analysis further gives the opportunity to quantify data of qualitative analyses [32]. Firstly, all transcripts were read by two researchers [SF, MD (MSc)] to gain a sense of the whole. Secondly, codes were generated independently by both researchers to capture all crucial contents, and to develop a codebook, if necessary, based on consensus decisions. We then screened the transcript of interview group 1 and assessed intercoder-reliability by applying the coding analysis toolkit CAT. We aimed at obtaining a Kappa value $\kappa \geq 0.61$ to ensure the developed codebook enhances reliability [33, 34]. Using the codebook, the other transcripts were analyzed while the initial coding scheme was continuously refined [35]. If necessary, codes were refined using joint discussions of the two researchers. As a result, a list of codes, i.e., a codebook, emerged representing reasons of choice regarding dental treatments. In a further qualitative analysis, (sub)categories on the reasons of choice were formed based on joint discussion between the researchers. Results of the CAT comparison tool, the coding-tree, the coding scheme including the codebook can be found in the S5 File, S2 Table.

2. We added a second analysis step and sorted statements according to the subject of choosing or not choosing dental treatments. If a statement could not be clearly assigned, it was initially defined as "unclear", then re-screened and (re)sorted according to the context of the transcripts. A first screening of the statements in interview group 1 was performed independently by the researchers (SF, MD). An intercoder-agreement of $r \geq 0.8$ was considered as acceptable [36]. After consensus decision was reached in a subsequent discussion, SF continued screening and sorting for the remaining interview groups. The number of sorted statements was counted and analyzed.

3. For analyzing assessment of the participants regarding cost coverage, statements with a focus on financial aspects were considered to get an in-depth understanding of consequences and challenges of additional patients' costs that may influence the choice for dental treatments.

## Results

### Interviews and participants' characteristics

Overall, 48 people registered and were invited for the interviews, ten participated in the pilot interview, and 27 participated in the focus group interviews 1–4. The remaining signed out at a short notice due to reasons of time or illness. We conducted four focus group interviews, with 5–9 participants each (which is in line with recommendations for focus groups [26]), that lasted 56–134 minutes. The participants were aged 22–74. As the point of data saturation was reached in interview group 2, we did not schedule more than the four planned appointments [37]. Individuals of all statuses regarding the participants' employment, education and income were represented. Most participants were female (n = 20) and SHI members (n = 26). More than half of them used a "Bonus booklet" (n = 16), some were enrolled in a "Bonus program" (n = 4) and had taken out a dental supplementary insurance (n = 6). Almost all participants had experienced out-of-pocket payments for dental treatments (n = 25). Descriptive information on the participants per focus group and in total are shown in Table 1.

### Results of the qualitative analysis: Reasons of choice regarding dental treatments from patients' perspective

We identified 53 reasons of choice for dental treatments divided in two categories (I) "health care service" and (II) "dentist & dental office", and four subcategories each (Fig 2). The supporting information contains detailed definitions for n = 24 reasons of the category "health care service" (S3 Table) and n = 29 reasons of the category "dentist & dental office" (S4 Table).

Additionally, a descriptive presentation of the number of reasons ranked as "most important" by the participants, a photograph of the chart board from group 3, and a descriptive collection and analysis of the reasons can be found in the S6–S8 Files.

**(1.I) Category "health care service".** The first category includes reasons of choice for dental treatments focusing on the service of dental care and its characteristics. It is further subdivided into subcategories "preconditions", "treatment", "costs", and "outcomes".

### Subcategory "preconditions"

One reason mentioned by the participants influencing their decision on dental treatments is *current complaints* (e.g., toothaches, visible dental problems), located in subcategory "preconditions": **"If I have a toothache or any noticeable or visible dental problems, then I am really [...] hurry to go to the dentist and get it treated. This is of course an absolute indication for visiting a dentist."(group 3, #8)**. Also, a perceived need for *prevention* leads to an appointment with the dentist. Furthermore, *patients' constitution* (e.g., overall health) may influence their treatment choice. *Professional recommendations* by the dentist and staff regarding a certain treatment (e.g., laser treatment, or dental crown) additionally affect our participants' decision-making. Indeed, the participants also reported the utilization of *self-diagnosis* based on information from different sources, e.g., family and friends, and the internet.

### Subcategory "treatment"

Further, *complaints during treatment process* can influence the participants' treatment choice. This reason refers, for example, to the treatment's invasiveness or pain experienced: **"[Laser]**

**Table 1. Descriptive information on participants per focus group and in total.**

| | Group 1 | Group 2 | Group 3 | Group 4 | Total |
|---|---|---|---|---|---|
| Number of participants | 7 | 6 | 9 | 5 | 27 |
| **Socio-demographic characteristics of participants (n = 27)** | | | | | |
| Age min-max in years (mean) | 23–66 (38.1) | 22–59 (48.2) | 22–76 (50.4) | 28–63 (41.8) | 22–76 (45.1) |
| Women/men | 6/1 | 5/1 | 5/4 | 4/1 | 20/7 |
| SHI/PHI | 6/1 | 6/0 | 9/0 | 5/0 | 26/1 |
| Employment status | | | | | |
| Fulltime employed | 3 | 3 | 2 | 1 | 9 |
| Part-time employed | 0 | 2 | 0 | 1 | 3 |
| Student | 3 | 1 | 2 | 1 | 7 |
| Unemployed | 0 | 0 | 1 | 0 | 1 |
| Pensions | 1 | 0 | 4 | 1 | 6 |
| Other | 0 | 0 | 0 | 1 | 1 |
| Education | | | | | |
| Graduation from university | 5 | 4 | 5 | 2 | 16 |
| Vocational training | 1 | 0 | 0 | 1 | 2 |
| Graduation after 13 school years (A-level) | 0 | 1 | 4 | 1 | 6 |
| Graduation after 10 school years | 1 | 1 | 0 | 1 | 3 |
| Net household income per month | | | | | |
| <750 | 1 | 1 | 2 | 0 | 4 |
| 750– <1,500 | 4 | 1 | 4 | 3 | 12 |
| 1,500– <2,500 | 2 | 0 | 0 | 2 | 4 |
| 2,500– <3,500 | 0 | 2 | 2 | 0 | 4 |
| >3,500 | 0 | 1 | 1 | 0 | 2 |
| Missing | 0 | 1 | 0 | 0 | 1 |
| **Use of incentive measures and supplementary insurance in dental care by participants (n = 26)** | | | | | |
| SHI—Using "Bonus booklet" | 2 | 6 | 5 | 3 | 16 |
| SHI—Using "Bonus program" | 1 | 2 | 0 | 1 | 4 |
| Supplementary dental insurance | 1 | 1 | 2 | 2 | 6 |
| **Experiences of participants with out-of-pocket payments (n = 27)** | | | | | |
| Already had to make out-of-pocket payments | 6 | 6 | 9 | 4 | 25 |

S/PHI—statutory/private health insurance.

is the most tooth-preserving method, the least painful method. And because it is a laser, practically no syringes are used. That means no anesthetics."(group 4, #2). Other reasons of choice focus on the *duration of treatment* such as resulting time efforts. Our participants declined a treatment that may last several hours (e.g., surgery, or root canal treatment), and prefer treatments with minimal time exposure (e.g., fast hardening tooth-filling).

## Subcategory "costs"

The participants also reported their personal need of receiving cost information about a planned treatment, which is reflected in the reason *actual costs*: **"If a treatment is necessary for me, then I need a guarantee whether I can bear it financially or if my insurance will cover it." (group 2, #2)**. Further reasons for decision-making include *out-of-pocket payments* and *income*. Our participants weighted up the *cost-benefit* ratio of a treatment and reported that having a *dental supplementary insurance* influenced their decision on a certain dental

**(I) Health care service**

Preconditions
current complaints,
prevention, patient's constitution,
professional recommendation,
self-diagnosis

Treatment
complaints during treatment process,
duration of treatment

Costs
actual costs, out-of-pocket payment,
income, cost-benefit, insurance coverage,
"Bonus booklet", "Bonus program",
dental supplementary insurance,
second offer, installment

Outcomes
aesthetics, durability, functionality,
influence on health, compatibility,
holism, complaints after treatment

**(II) Dentist & dental office**

Professional skills
training, work experience, medical error,
accuracy, calmness, flexibility,
interdisciplinarity, professional treatment and
costs information, adequate advice,
language barrier

Social skills
interhuman relations, trust,
dentist takes time, courtesy/friendliness,
ability to take criticism, patient opinion,
profit orientation, obtrusiveness,
presentation of alternatives, seriousness,
appearance

Office staff & equipment
medical staff (not dentist),
non-medical staff,
medical-technical equipment,
non-medical equipment

Office processes
patient orientation, coordination,
waiting time, hygiene

Legend: (I)/(II) categories, subcategories, reasons. Order of reasons do not imply a weighting.

**Fig 2. Reasons of choice regarding dental treatments from patients' perspective classified in (sub)categories.**

treatment. Beyond, the *"Bonus booklet"* and a *"Bonus program"* were mentioned. Both incentive measures result in financial bonuses limiting patients' out-of-pocket payments. The participants also reported to use a *second offer* by other dentists to reduce costs, and the opportunity of *installments*.

### Subcategory "outcomes"

The participants' choice is also led by treatments' outcomes. Reasons of category "outcomes" are *aesthetics*, *compatibility*, *durability*, *functionality*, and *influence on health*: **"It has been**

proofed for a long time that dental health correlates directly with lifetime. **Healthy teeth, long life. Teeth reflect health of various internal organs and so on." (group 4, #2)**. They also mentioned preferred approaches such as *holism* (containing biological methods). The participants further expected to leave the dental office with no *complaints after treatment*.

(1.II) Category "dentist & dental office". The second category includes reasons of choice for dental treatments that focus on the professionals performing dental treatments as well as office structures and processes. Subcategories include "professional skills" and "social skills" of dentists as well as "office staff & equipment", and "office processes".

## Subcategory "professional skills"

Reasons of subcategory "professional skills" include the dentist's *training* and *work experience*. Our participants avoid a treatment in case they experienced *medical errors* beforehand. They expect *accuracy* in the dentists' work in terms of observing physiological (e.g., detection via radiograph) and psychological conditions (e.g., asking for current live situation). Furthermore, *calmness* (e.g., relaxed or choleric behavior) and *flexibility* (e.g., reacting on dental fear with rescheduling an appointment) play a role in decision-making. Additionally, the participants look for *interdisciplinarity* in the dental office, in order to have a broad basis of information. They reported to expect a *professional treatment and costs information*: **"There is never only one way in life. This assumption does not exist, and I think the dentist is obliged to show and explain several options to me. And if it is cost-intensive, then I do have the right to know what I'm paying for." (group 2, #4)**. Furthermore, the participants prefer *adequate advice* in terms of receiving understandable and relevant information. According to their experiences *language barriers* can lead to misunderstandings regarding their wishes and needs.

## Subcategory "social skills"

Reasons of subcategory "social skills" depict the dentist's ability to establish *interhuman relations* and *trust*: **"I have been with the current dentist for over 20 years now. I go there regularly. I know that my dentist recommends those things that are necessary, but not unnecessarily things that go beyond what I need. So, I have a lot of trust in her." (group 3, #1)**. The participants also appreciate when the *dentist takes time* (e.g., accompanying patient to appointment desk). *Courtesy* and *friendliness* of dentist and staff is another criterion, leading to a pleasant atmosphere. Furthermore, the participants expect the dentist's *ability to take criticism* and involvement of *patients' opinion* in treatment decisions. Our participants reported they might decide against a treatment when assuming dentist's *profit orientation*. They criticized *obtrusiveness* of dentists (e.g., little time to think about treatment opportunities), wish for *presentation of alternatives*, and expect *seriousness*. Furthermore, the dentist's *appearance* was mentioned to play a role in deciding for or against dental treatments.

## Subcategory "office staff & equipment"

Reasons of subcategory "office staff & equipment" include, for instance, a sufficient number of employees and the appearance (e.g., good-looking) of both *non-medical* and *medical staff*. Regarding the equipment, our participants prefer innovative, not worn-out dental apparatus and appreciate comforts in waiting rooms, depicted by the reasons of choice *non-medical equipment*, and *medical-technical equipment*: **"The most modern equipment is important for me. And of course, it is a problem when you go to a dentist, who has equipment that is 20 years old, and he is not in the mood to take out a loan. In the end the dental treatments look like that." (group 4, #2)**.

### Subcategory "office processes"

Different reasons can be found in subcategory "office processes". For example, *patient orientation* has mentioned as crucial factor in decision-making by the participants. Also, they valued appropriate *coordination* of services after experiencing deficiencies (e.g., missed appointment by dentist): **"What I expect from my dental office is, that if I have an acute problem, I can go there directly whatever happens, and they will take care of me." (group 3, #1)**. Furthermore, our participants considered the reason *waiting time* and measures of *hygiene*. Further statements for reasons can be found in the S5 Table.

**(2) Consideration of reasons for choosing or not choosing dental treatments.** The results of sorting the statements into "reasons for choosing" and "reasons for not choosing" dental treatments show that most reasons (n = 47) are ambivalent, i.e., reasons trigger decisions for and against a treatment. About two thirds of these reasons (n = 32) contain statements that are predominantly (>50%) used when choosing treatments (e.g., out-of-pocket payment, patient's constitution). A smaller proportion (n = 16) contains statements that are mostly (>50%) used against choosing treatments (e.g., aesthetics). For n = 5 reasons, statements are distributed equally (e.g., durability). Some reasons contain statements that could not be clearly assigned to either the "*choosing dental treatments*" or "*not choosing dental treatments*" group and remain "unclear". Only n = 4 reasons were given solely "for choosing dental treatments" ("Bonus booklet", "Bonus program", non-medical staff, ability to take criticism) and n = 3 for "not choosing dental treatments" (adequate advice, medical error, coordination). Calculation of intercoder-agreements can be found in the S9 and S10 Files. S6 Table and S11 File give an overview on the distributions of sorted statements and reasons, schematic and descriptive, including the analysis.

**(3) Cost coverage for dental treatments from patients' perspective.** The participants reported, dental treatments such as implants sometimes can be very cost intensive. Some participants would decide for a treatment with a high out-of-pocket payment and therefore quit another comfort such as vacation. Others would decline a treatment due to high costs. Burden of out-of-pocket payments is affected heavily by *income*, *cost-benefit* ratio, and *actual costs* of a treatment. The participants explained that their current employment status plays an important role such as being a student. Furthermore, they weighted benefits against costs when deciding between treatment options. Realistic costs estimations are challenging when deciding for or against dental treatments. However, there are opportunities used by the participants to lower costs. They actively manage their financial burden of dental treatments by measures offered through health insurances, e.g., "Bonus booklet" and "Bonus program", and self-initiated measures such as requesting a second offer, taking out a dental supplementary insurance, and using installments (Table 2).

## Discussion

While there have been studies focusing on patients' decision-making with regard to certain dental treatments or subgroups, the key strength of this study is that we captured all adult patient groups. The diversity is strengthened by the fact that the study focused on identifying and understanding patients' reasons of choice for or against dental treatments in general. In addition to the previously known reasons for choosing dental treatments such as out-of-pocket payments and trust in the dentist, new reasons emerged, e.g., holistic treatment. The reporting quality of the study has been ensured by applying the instrument COREQ.

This study provides detailed insights to our participants' reasons for (not) choosing dental treatments and its financing. We identified 53 reasons split into two main categories (dental health service, dentist & dental office) and further divided into eight subcategories. Our

**Table 2. Challenges and opportunities of financing dental care from patients' perspective, and statements.**

| Challenges and opportunities from patients' perspective | | | Statements |
|---|---|---|---|
| Challenges | Actual costs and estimation | | You never know what it will cost in the end, at least not exactly. (group 1, #1) // My dentist said, he thinks, that we shouldn't try dental crowns. Because it just costs a lot of money and you never know what will happen in five years, or how it feels after one year. (group 1, #1) |
| | Individual income | | Unfortunately, I broke my bite splint. I haven't worn it since then because I am a student. [. . .] I haven't done it yet because you must pay for it yourself. (group 1, #6) // I am currently in a part-time job, and I also have a severe disability. For me, the financial aspect matters a lot. (group 2, #2) |
| | Cost-benefit estimation | | I would probably first ask my family if someone had any experience and then also search online and weight all options. (group 1, #6) // I was then given the choice "Okay, let's do a plastic filling now, but you have to expect that it must be replaced after a few years. Or should we do a ceramic filling that would somehow cost a 500, 600?". And that would be ok, probably, because it would be more reasonable. But that's a decent price tag. (group 3, #2) |
| Opportunities | Instruments of health insurance | "Bonus booklet" | Once a year, I have a dental check-up, and get it signed within my "Bonus booklet". I can prove it since the 90ies, that I have done it. I really take care of it. (group 3, #9) // The "Bonus booklet" is very important for me, because then I am always up-to-date and get my stamp every year. If a denture really is on agenda, I try to keep my out-of-pocket payment as low as possible. (group 3, #8) |
| | | "Bonus program" | When you have a lot of stamps, because you have visited a lot of different physicians, e.g., you got a vaccine, then you get money back. [. . .] Of course, I run to my physicians and get my stamps. (group 2, #6) |
| | Further measures | Dental supplementary insurance | I pay only 50%, because implants cost an arm and a leg. (group 2, #5) // Then I took out a dental supplementary insurance, because it was extremely expensive for me [. . .] those braces and stuff. Otherwise, you always must pay for these things. [. . .] It can be considered as a subscription or something similar you monthly pay for [. . .] and when you need a larger amount at some point you don't have to pay it all at once. (group 4, #1) |
| | | Second offer | I got a cost calculation from my dentist and they said that there is a website where you can enter data and dentists will get in touch with you and make an offer. And instead of 3,000, different dentists offered 2,000 and 1,000. (group 2, #4) // What I have experienced via the website "Second dentist's opinion" [. . .], I got some offers for the same treatment [. . .] with differences up to 2,000. (group 4, #4) |
| | | Installment | I wanted to have a fixed prosthesis and it was medically possible. And this dentist gave me, because that treatment was not covered by my health insurance, the opportunity to pay a defined amount every month. So, that was an arrangement between us, even though it was a lot of money. (group 3, #5) |

(no. of interview group, participant #); // next statement.

findings correspond to the results of previous studies, undertaken in other countries, presenting reasons such as language barriers [38], complexity of treatments [39] and time of treatment [12, 13]. The participants in our study assessed out-of-pocket payments as a most important reason for decision-making. They estimated dental treatments as "costing an arm and a leg" (group 2, #5) and concluded that they had to choose between "you go twice a year on vacation [..] [or] I have teeth in my mouth" (group 2, #5). Although, this finding might not be representative for Germany, the reason was also identified in several international studies [12, 38, 40]. Coverage of dental health care by statutory health insurances plays a major role in this context. Although German SHI patients have small co-payments compared to most European countries and a broad benefit package, out-of-pocket payments arise, for instance, regarding dentures [6]. Those are perceived as high by the participants. "Costs are of course a reason to say 'I'll postpone the treatment', 'I'll cancel it', or 'I won't do it at all'." (group 1, #1). Using incentive measures, SHI pursues the goal to increase efficiency of offered health services [41]. The approach has been proved as efficient for health insurances [42, 43], and "very important" (group 3, #8) by the participants. A study showed, that patients using a „Bonus booklet"had a better dental status than those without [24]. Another study found that the presence of a „Bonus booklet"promotes dental visits [44]. This is in line with statements of our participants,

who state that they would "go dutifully to these annual check-ups for receiving stamps" (group 2, #4). For the participants, the instrument leads to the desired success of costs reduction if check-ups are used regularly. However, "going to the dentist is [still] expensive" (group 1, #5), especially "implants might be very costly" (group 3, #7). Further measures considered by our participants, they are "very glad to have" (group 2, #5), include dental supplementary insurances since they pay "only 50% for an implant" (group 2, #5). In 2017, a total of 15.7 million patients in Germany had dental supplementary insurances, with an upward trend [45]. Internationally private insurances might play a more important role in dental care. For example, in the Netherlands, more than 80% of the population takes out supplementary insurances, that cover dental and other health services [6]. This clearly shows the difference in the benefit basket for dental services between the countries. Additionally, our participants reported, they would use the opportunity "that it is enshrined in law that every patient has the right to a second or even a third opinion" (group 3, #7) on treatments and costs from another dentist. They are led by experiences of previous second opinion consultations, when the dentist said ‴no, that [treatment] is not necessary‴. (group 2, #3), or when they got "different offers for the same treatment, and there were differences of up to 2,000." (group 4, #4). A further approach for a reduction of costs is seen in dental tourism, although not widely used by German patients and not mentioned by our participants. Patients of countries such as Austria or the UK more often use this opportunity. Some countries developed a medical tourism industry offering a variety of medical and dental treatments [46, 47]. The participants try to contrive ways to reduce their out-of-pocket costs. However, the non-use of a treatment may lead to serious medical [1] and social consequences [2, 3]. In the long run, limited financing of dental treatments might be more expensive compared to a broader coverage. Therefore, the influence of out-of-pocket payments should be investigated quantitatively to determine patients' willingness-to-pay for dental treatments. A so far unknown reason of choice for dental treatments from patients' perspective includes the desire for holistic and biological approaches. Those treatments of complementary and alternative medicine (short: CAM) include, for instance, acupuncture and herbal medication. The participants would select CAM "as a priority" (group 4, #5), because they "just don't know, whether [a substance] has a 100% antitoxic effect, is degraded, or stays as a residue [in the body]." (group 4, #1). Also, an explorative, cross-sectional survey among practicing dentists showed that CAM is offered by several dentists and requested by patients [48]. However, the evidence regarding effectiveness and safety of these treatments is often scarce and unclear [48, 49]. CAM treatment costs must be paid fully by patients in Germany since it has not been included in the SHI benefit basket.

Besides "health care service", our participants also addressed reasons that can be attributed to the "dentist & dental office". In this regard, dentists' training status is considered important by the participants because "a dentist who has a training that goes back 30 years and has not undergone any further training himself [...] cannot offer a modern care and has no benefit" (group 4, #2) for them. Furthermore, a relationship of trust between patient and dentist plays a major role for our participants, when reporting being "with the current dentist for over 20 years [...] [, knowing] that [the] dentist recommends those things that are necessary, but not unnecessarily things [...] [, and consequently to] [...] have a lot of trust in [the dentist]" (group 3, #1). Obviously, professional and social skills are relevant in decision-making for them. This was also previously shown in studies for Brazil [50], and Sweden [51]. Content of subcategory "social skills" was named most often as "most important" by the participants over all subcategories, including reasons such as the dentist's ability to take criticism and accepting patient's opinion. A personal relationship between patient and dentist is based on mutual sympathy, but unlike coverage decisions of dental treatments, it cannot be regulated by law. Dentists in Germany are committed to periodically participate in trainings on technical and social

contents [52]. However, there might be a need of extending education on social competences. The participants further reported using the internet as source of knowledge to receive information on how "risky [a dental treatment] can be" (group 1, #5), or on "compatibility of [dental] materials" (group 1, #6). However, as reported in earlier studies, patients found internet information often to be incomplete, since contents might be of moderate quality and data might not be reliable [53, 54]. Professional directing from dentists regarding treatments is necessary to produce best outcomes [55]. Furthermore, adequate consultations are necessary to save patients from medically wrong decision-making that may result in severe health consequences [56]. Internet information should be revised by dental experts. Online portals with an assessment of benefits and risks of dental services in general can support patients in their decision for or against a specific treatment. Evaluation portals for hospitals already exist [57]. Overall, more attention should be paid to patients to support shared decision-making processes on dental treatments.

The difference between "reasons for choosing" and "reasons for not choosing" dental treatments showed that only a few reasons alone condition a decision in one direction. Financial incentives measures such as the "Bonus booklet" and "Bonus programs" always led to a positive decision for dental treatments by our participants. The effects of those incentives are well known. Studies show, that adequately designed incentives lead to positive treatment decisions [58, 59]. Most of the reasons led to both decisions for and against dental treatments. The participants' statements are decisive for as well as against the choice in this regard. Some reasons might result exclusively from patients' preferences and their personal characteristics. For instance, aesthetics is perceived differently by each patient [60, 61]. Compatibility is also patient-specific, e.g., due to allergies [62]. Other reasons do not allow for a clear assessment, because it can take different values (e.g., low out-of-pocket payments) [10].

The findings of this study must be considered in the light of limitations linked to the methods, and to the recruiting process that is restricted to specific channels and one region. Although, focus group interviews have the opportunity of thought-provoking impulses within a discussion, this can also lead to a restrained behavior of participants. They may not express their overall opinions. Due to our recruitment process selection bias might have been occurred [63]. In addition, because of the small number of participants, which is in the nature of the method, the sample of our participants is not representative for the German population [64]. Furthermore, our approach of purposive sampling excludes the claim of representativity; however, it enables us to gain "information-rich" cases with experience and interest in the topic [65]. We indicate that certain behaviors concerning the utilization of dental treatments (e.g., oral health behavior, decision-making, or willingness-to-pay) may differ between regions. Similarly, participants with a university degree were above the German average while unemployed participants were slightly below [66]. Most participants in many dental qualitative studies are female, like in our study [67]. Also, reflexivity of the researchers should be briefly assessed. It can be confirmed that all researchers have already had experience as patients in dental care. Due to research and teaching activities, SF and CH also had knowledge of the German healthcare system including dental care. This, and experiences in the course of the study, could have had an influence on implementing and analyzing the interviews [68, 69].

## Conclusions

Our participants reported to have various reasons for choosing or not choosing dental treatments (e.g., current complaints, cost-benefit). Certain reasons are named more important by the participants, e.g., out-of-pocket payment. Although SHI widely covers costs of dental services in Germany compared to many other countries, the participants assess cost coverage as

insufficient for certain treatments. As a result, they look for cost-reducing measures or decide against medically necessary treatments. Consequently, costs of dental services can be a barrier to access dental care leading to additional disease related and financial burden for patients and the healthcare system. To prevent cost-intensive dental treatments, health insurances should strengthen incentive measures for preventive services as those are crucial especially for oral health. Patients should be sensitized to use existing SHI preventive measures at an early stage and be supported in making decisions about dental treatments. To ensure communication between patient and dentist, dentists should be trained in special programs fostering shared decision-making. Patients then have the opportunity to make their dental treatment decisions based on expert information (e.g., risks and benefits of a treatment). As the results must be interpreted in the light of the qualitative study design, further studies should quantify the relevance of the reasons for choosing dental treatments from patients' perspective (e.g., in a willingness-to-pay analysis). Based on this, identified reasons can be considered in future policies to improve patients' utilization behavior (e.g., improved information sources, increased incentive measures).

## Supporting information

**S1 File. COREQ (COnsolidated criteria for REporting Qualitative research) checklist.**
(PDF)

**S2 File. Participants' information document.**
(PDF)

**S3 File. Participants questionnaire on socioeconomic characteristics.**
(PDF)

**S4 File. Informed consent.**
(PDF)

**S5 File. Results of the CAT comparison tool.**
(XLSX)

**S6 File. Descriptive presentation of "most important" reasons.**
(TIFF)

**S7 File. Photograph of chart board selecting "most important" reasons in interview group 3.**
(TIFF)

**S8 File. Descriptive collection and analysis of "most important" reasons.**
(XLSX)

**S9 File. Calculation of intercoder-agreement sorting "unclear" statements.**
(XLSX)

**S10 File. Calculation of intercoder-agreement sorting statements into reasons for (NOT) choosing.**
(XLSX)

**S11 File. Overview and analysis of distributions of sorted statements on reasons.**
(XLSX)

**S1 Table. Interview guide used in focus groups.**
(DOCX)

**S2 Table. Coding-tree and coding scheme, including codebook.**
(DOCX)

**S3 Table. Definitions of category "health care service", its subcategories and reasons (n = 24, no.1–24).**
(DOCX)

**S4 Table. Definitions of category "dentist & dental office", its subcategories and reasons (n = 29, no.25–53).**
(DOCX)

**S5 Table. Additional statements on patients' reasons for (not) choosing dental treatments.**
(DOCX)

**S6 Table. Distributions of sorted statements on reasons.**
(DOCX)

## Acknowledgments

We thank all participants who took part in the interviews for their time and sharing their experiences in dental care and opinions with us. Also, we are grateful to illustrator Ms. Juliane Lüke providing an image template for Fig 2 and for the support of Mr. Fabian Roelen (FR) as student assistant. The authors have no conflict of interest to declare related to the content of the study.

## Author Contributions

**Conceptualization:** Susanne Felgner, Cornelia Henschke.

**Data curation:** Susanne Felgner, Marie Dreger.

**Formal analysis:** Susanne Felgner, Marie Dreger.

**Funding acquisition:** Cornelia Henschke.

**Investigation:** Susanne Felgner, Cornelia Henschke.

**Methodology:** Susanne Felgner, Marie Dreger.

**Project administration:** Cornelia Henschke.

**Software:** Susanne Felgner, Marie Dreger.

**Supervision:** Cornelia Henschke.

**Validation:** Cornelia Henschke.

**Visualization:** Susanne Felgner, Cornelia Henschke.

**Writing – original draft:** Susanne Felgner.

**Writing – review & editing:** Marie Dreger, Cornelia Henschke.

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
