## [Decision Letter · Decision Letter 0]

12 Jan 2021

PONE-D-20-35002

Factors of choice in dental care treatment – A qualitative study based on patients’ perspective

PLOS ONE

Dear Dr. Felgner,

Thank you for submitting your manuscript to PLOS ONE. After careful consideration, we feel that it has merit but does not fully meet PLOS ONE’s publication criteria as it currently stands. Therefore, we invite you to submit a revised version of the manuscript that addresses the points raised during the review process.

Thank you for submitting your research in the field of Dentistry to PLOS One. You have chosen a crucial point in the field of oral health services research and thereby I invited experienced reviewers coping with this subject.

Furthermore, I am also involved in research in this field.

With regard to the reviews as well as the publication guidelines of PLOS ONE there are two possibilities for your manuscript:

a) a major revision

b) a retraction to hand it in somewhere else.

In case of a) you and your COs have to rework the data and rewrite the paper completely regarding

the points raised by both reviewers. This encompasses a clear explanation of the

limitations due to participant selection (bias) of your study by age and location (Berlin is not representative

for Germany. Patients in Stuttgart area are different in their oral health behavior, decision and willingness to

pay compared to Berlin)

You should consult more literature and a dentist with a German license to get more insight to the system.

For instance, "out of the pocket" payments are applicable for treatments which are not

standard of the SHI (BEMA catalog) and in case of dentures for patients not

in social support. Last mentioned will get the SHI-based denture free of charge.

Sometimes these options might not be offered to the patients due to cost-benefit-calculations

of the dental practice - which is a violation of law. Some of these aspects can be found i.e. in Behrend & Huettig 2016

as well as findings from patient interviews in Florian M. Behrends thesis (available as PDF via Uni Tübingen).

You can find more literature there - even if the amount is limited for Germany still today.

In case of your decision for a) I am looking forward to handle your revision.

We look forward to receiving your revised manuscript.

Kind regards,

Fabian Huettig, DMD, Ph.D.

Academic Editor

PLOS ONE

Journal Requirements:

2. When reporting the results of qualitative research, we suggest consulting the COREQ guidelines: http://intqhc.oxfordjournals.org/content/19/6/349

In this case, please consider including more information on the interviewers, their training and characteristics; and please provide the interview guide used.

Reviewers' comments:

Reviewer's Responses to Questions

**Comments to the Author**

1. Is the manuscript technically sound, and do the data support the conclusions?

Reviewer #1: Partly

Reviewer #2: Partly

2. Has the statistical analysis been performed appropriately and rigorously? 

Reviewer #1: Yes

Reviewer #2: N/A

3. Have the authors made all data underlying the findings in their manuscript fully available?

Reviewer #1: Yes

Reviewer #2: No

4. Is the manuscript presented in an intelligible fashion and written in standard English?

Reviewer #1: Yes

Reviewer #2: No

5. Review Comments to the Author

Reviewer #1: The authors present a study about reasons of choice in dental care treatment. In Germany, every adult visits the dentist 2.4 times a year on average. In addition to paid preventive care, there are four reasons for this: pain, functional limitations, appearance and psychosocial impairments. Many older people have dentures (that is good).

Sometimes over-treatment takes place (that is bad). For some groups in the population, however, there are barriers to using dental care: children, the very old, the disabled, multimorbid (ASA 3+), pain or fear patients etc. For these groups we need qualitative studies in order to break down the barriers but not for healthy middle-aged women to increase the risk of over-treatment. The topic and the methodology of the paper are ok, only the target group is unsuitable in the opinion of the reviewer.

Reviewer #2: 1 Summary of the research

The main research question of the current manuscript is twofold: (a) What factors influence patients in Germany in their decision on dental treatment? (b) How do patients assess cost coverage of dental services by German statutory health insurance? The authors claim to have identified (a) themes affecting patients’ decision-making on dental treatment and (b) financial challenges of dental treatments for the patients and ways how they take action against them. The authors draw the following conclusions: There are different reasons for choosing or not choosing dental treatment which can be categorised hierarchically into two main categories with several sub-categories each. Furthermore, financial aspects play a major role in decision-making for or against dental treatment and patients employ different measures to alleviate this financial burden.

One strength of this study is that the research question is probably of interest for many different audiences, e. g. researches, practitioners, policy makers, and patients alike. Moreover, the approach is suitable for identifying (until today) unknown reasons for (not) choosing dental treatment. Finally it is to be welcomed that the authors used member checking as a quality assurance measure, i. e. they asked the participants after a break to confirm, reject, and complete the identified reasons in order to obtain a “peer-approved” list of reasons.

But the research questions also bear weaknesses: If these questions would have been formulated in a more concrete manner more valuable insights could have been revealed. Further, the authors made not full use of the strengths of the qualitative methods employed. As a consequence the results remain needlessly superficial: It seems highly likely that new insights can be revealed from the data if additional data analysis steps would be taken.

On the whole I recommend to accept the manuscript for publication after a major revision.

2 Examples and evidence

2.1 Major issues

There is one major issue which should be addressed by the authors primarily. Before I begin with the major issue I present two minor issues because all three of them are intertwined. Further minor issues follow in the next section.

I suggest changing the manuscript title from “factors of choice in dental care treatment” to “reasons for (not) choosing dental care treatment”. “Factors of choice” seems misleading since (a) it could be understood as “factors which should be chosen”; (b) it insinuates a quantitative approach instead of a qualitative one: In the manuscript the researchers do not analyse causal factors derived statistically but subjective reasons which people give for their actions.

I suggest changing the first research question from “What factors influence patients in Germany in their decision on dental treatment?” to “What reasons do patients in Germany give for choosing or not choosing dental treatment?”. Mentioned themes found in transcripts are interesting per se but the elephant in the room is the question whether these themes represent reasons in favour of or against health care utilisation. I suggest changing the second research question from “How do patients assess cost coverage of dental services by German statutory health insurance?” to “What do patients think about cost coverage of dental services by German statutory health insurance?”. The term “assess” is ambiguous: It could mean “how do they think about cost coverage” (in the sense of “associations/ideas”) or it could mean “how do they evaluate cost coverage” (in the sense of “good/bad”). The reported results imply that the former interpretation seems adequate.

One major issue arises from the data analysis. The manuscript text in the results section indicates that the focus group participants did not only name themes which revolve around dental health care utilisation but also whether the mentioned aspects represent reasons for choosing or not choosing dental treatment. It seems to suggest itself that this crucial dichotomy should have been coded and analysed as well: Firstly, I propose to use the additional codes “reason for choosing dental treatment” and “reason for NOT choosing dental treatment”. Secondly, it should be analysed which of the already identified reasons occur mainly in conjunction with the code “reason for choosing dental treatment” and with the code “reason for NOT choosing dental treatment” – and which reasons seem to be ambivalent because some people give these reasons for choosing dental treatment and others for NOT choosing dental treatment. It would be heuristically instructive if the authors would visualise the results like in figure 2 including the categories “group of themes mainly interpreted as reasons for choosing dental treatment” and “group of themes mainly interpreted as reasons for NOT choosing dental treatment” and a third category “group of ambivalent themes, sometimes interpreted in favour of and sometimes against dental treatment”.

This is an important issue: In the realm of quantitative research it would be like having large-scale survey data suited for multivariate inferential statistical analysis and analysing only descriptive frequency distributions and cross-tabulations.

2.2 Minor issues

In general, “factors of choice in dental treatment” should be replaced by “reasons for (not) choosing dental treatment”.

Is the term “performance” the adequate term for the subcategories and codes which this category contains? “Health care service” seems to comprise the meanings better.

The detailed excursus on the coverage of dentures in Germany should be explicitly framed as additional information so readers have more context to understand the results (if that is its purpose). Otherwise this excursus does not seem relevant for the manuscript since neither the data generation process (focus groups) nor the data analysis process nor the results focus on dentures.

The methods should be explained in more detail, to make the way of the data more comprehensible, which steps were taken and so on. The authors should take heed of the “Standards for Reporting Qualitative Research: A Synthesis of Recommendations” by O’Brien et al. (2014). Especially, (a) they should reflect more on the sampling strategy: Why were participants recruited that way and what are the implications of this approach? (b) For the readers the main questions from the manual for the focus groups are necessary to gain a better grasp of the data generation process. (c) Relevant excerpts from the codebook or coding-scheme could be shown to retrace the data analysis process.

Data availability is limited due to participant privacy. This is a common issue with qualitative data but the authors offer access to anonymised interview transcripts on demand.

Finally, further proof-reading is necessary: There are still some orthographical and grammatical errors left.

6. PLOS authors have the option to publish the peer review history of their article (what does this mean?). If published, this will include your full peer review and any attached files.

Reviewer #1: No

Reviewer #2: No

---

## [Author Response · Author response to Decision Letter 0]

29 Mar 2021

Dear Dr. Hüttig,

First and foremost, we would like to thank you and your team for providing us with the opportunity to revise the manuscript. We have taken care to consider the reviewers’ concerns carefully, addressing each in as straightforward a manner as possible. We found their comments to be insightful and helpful and feel that the manuscript has benefited greatly as a result.

Below please find the responds to each point raised by the academic editor and both reviewers. 

We included a marked-up copy of our manuscript that highlights changes made to the original version 'Revised Manuscript with Track Changes' as well as an unmarked version of our revised manuscript ('Manuscript'). We have also added further documents as supporting information.

Please do not hesitate to contact us should anything remain unclear, or if any further revisions are needed. 

Thank you, again, for your time and consideration!

Sincerely,

Susanne Felgner

Editor’s comments

“In case of a) ([a major revision]) you and your COs have to rework the data and rewrite the paper completely regarding the points raised by both reviewers. This encompasses a clear explanation of the limitations due to participant selection (bias) of your study by age and location (Berlin is not representative for Germany. Patients in Stuttgart area are different in their oral health behavior, decision, and willingness to pay compared to Berlin). You should consult more literature and a dentist with a German license to get more insight to the system. For instance, "out of the pocket" payments are applicable for treatments which are not standard of the SHI (BEMA catalog) and in case of dentures for patients not in social support. Last mentioned will get the SHI-based denture free of charge. Sometimes these options might not be offered to the patients due to cost-benefit-calculations of the dental practice - which is a violation of law. Some of these aspects can be found i.e. in Behrend & Huettig 2016 as well as findings from patient interviews in Florian M. Behrends thesis (available as PDF via Uni Tübingen). You can find more literature there - even if the amount is limited for Germany still today.”

Thank you very much for your helpful comments and suggestions. We reworked the data and rewrote huge parts of the paper according to the points raised by the reviewers. We additionally analyzed the data according to the reviewer’s (#2) suggestions.

We fully agree that the limitations regarding the selection of participants need to be clearly explained in the manuscript. For our study, we exclusively recruited participants aged ≥ 18. For organizational reasons, we conducted the interviews at the Technische Universität Berlin. Therefore, the recruitment strategy was limited to Berlin locations and media, except for the advertisement on Facebook and eBay. Being the study’s analyzing researchers, we do not know the actual participants’ residence. We also did not request the postal codes via the questionnaire that participants filled in. The limitations section now includes a more critical view on representativeness of the results. We indicate that certain behaviors concerning dental treatments (e.g., oral health behavior, decision-making, or willingness-to-pay) can be region-specific and may lead to bias. 

We assumed we had clearly presented the regulations of co-payment within the framework of the SHI-system, with a focus on dentures (e.g., “Although many basic treatments such as dental check-ups and tooth-fillings [16] are covered fully by SHI, treatments exceeding standard care services as well as dentures are only covered in parts [17].”; p.4 in the manuscript). Of course, we made revisions and additions to the manuscript that contribute to readers’ understanding. For example, we now clearly explain the exception mentioned in the Social Code Book V by adding information on up to 100%-coverage for patients fulfilling certain criteria (e.g., patients with low income, or receiving welfare beneficiaries; p.5 in the manuscript). We also added an arrow to Fig 1 representing the scope of SHI-payment. This arrow depicts the 100% SHI-payment for dentures. 

 

Journal Requirements

“1. Please ensure that your manuscript meets PLOS ONE's style requirements, including those for file naming. The PLOS ONE style templates can be found at https://journals.plos.org/plosone/s/file?id=wjVg/PLOSOne_formatting_sample_main_body.pdf and https://journals.plos.org/plosone/s/file?id=ba62/PLOSOne_formatting_ sample_title_authors_affiliations.pdf.” 

According to the PLOS ONE style templates, we changed the file naming of figures (e.g., “Fig1.tiff”) and the supporting information (e.g., “S1_Table.docx”). We also changed the citation of figures (e.g., “Fig 1”), tables (e.g., “Table 1”) and supporting information (e.g., “S1 Table”) in the manuscript, and the table titles in the supporting information documents. The article title is now written in non-bold letters.

“2. When reporting the results of qualitative research, we suggest consulting the COREQ guidelines: http://intqhc.oxfordjournals.org/content/19/6/349. In this case, please consider including more information on the interviewers, their training and characteristics; and please provide the interview guide used.”

We now follow the criteria of the COREQ checklist. Accordingly, we added information to respective sections (e.g., methods section). We also used the COREQ checklist for explicit and comprehensive reporting of our qualitative study. We added the table to the supporting information (S1 File). In this respect, we have taken the opportunity to give more information about the interviewers/researchers. For example, we added additional information of the interviewers (SF, CH) on their experiences in conducting and analyzing qualitative studies [e.g., “Provision and financing of assistive technology devices in Germany: A bureaucratic odyssey? The case of amyotrophic lateral sclerosis and Duchenne muscular dystrophy.” by Henschke (2012), and “Physicians' Decision Making on Adoption of New Technologies and Role of Coverage with Evidence Development: A Qualitative Study.” by Felgner, Ex & Henschke (2018)]. In addition, SF and CH have been teaching qualitative research methods at the university. Researcher MB was also involved in the interviews as an assistant (responsible for recording). Later she also analyzed the interviews as employed research fellow. In the methods section, we added characteristics of the moderator, co-moderator, and assistants, e.g., sex, and positions at the department. Further, we added the interview guide to the supporting information (S1 Table). 

“3. We note that you have indicated that data from this study are available upon request. PLOS only allows data to be available upon request if there are legal or ethical restrictions on sharing data publicly. For information on unacceptable data access restrictions, please see http://journals.plos.org/plosone/s/data-availability#loc-unacceptable-data-access-restrictions. In your revised cover letter, please address the following prompts: 

b) If there are no restrictions, please upload the minimal anonymized data set necessary to replicate your study findings as either Supporting Information files or to a stable, public repository and provide us with the relevant URLs, DOIs, or accession numbers. Please see http://www.bmj.com/content/340/bmj.c181.long for guidelines on how to de-identify and prepare clinical data for publication. For a list of acceptable repositories, please see http://journals.plos.org/plosone/s/data-availability#loc-recommended-repositories. We will update your Data Availability statement on your behalf to reflect the information you provide.”

Thank you for calling this important point to our attention. Before conducting the interviews, an ethics application was submitted to, and approved by the ethical committee of the Technische Universität Berlin. The application addresses the use of collected data: by signing the informed consent, interview participants confirmed that anonymized quotes based on the anonymized transcripts could be used for publication as part of the research project. In compliance with this requirement, we give single statements of the interview participants in the manuscript (Table 2, p.13; Fig 2) and the supporting information (S8, S9 and S10 Files, and S5 Table) for traceability and support of our study findings. We do not offer access to anonymized interview transcripts in another format, extend, or on demand. We have updated our Data Availability statement regarding this information. 

According to the informed consent personal data of the participants is only presented in aggregated form so that no conclusions can be drawn about individual persons. For this reason, we do not provide the data set on the results of the survey on socio-demographic characteristics by questionnaire.

We have also attached the participants’ information document, the informed consent, and the participants questionnaire on socioeconomic characteristics as supporting information to the manuscript (S2, S3, and S4 Files).

Contact information of the ethical committee is: 

Ethical committee of the Department of Psychology and Ergonomics, Technische Universität Berlin; Secretary MAR 3-2, Marchstraße 23, 10587 Berlin, Germany; phone: 049/03031422370, email: ethik@ipa.tu-berlin.de, chairman: Dr. Stefan Brandenburg. 

Reviewers’ comments

“1. Is the manuscript technically sound, and do the data support the conclusions? The manuscript must describe a technically sound piece of scientific research with data that supports the conclusions. Experiments must have been conducted rigorously, with appropriate controls, replication, and sample sizes. The conclusions must be drawn appropriately based on the data presented. Reviewer #1: Partly[,] Reviewer #2: Partly”

Thank you very much for the assessment. We hope to meet the mentioned criteria even better after revising and making additions to the manuscript according to the following comments. 

“2. Has the statistical analysis been performed appropriately and rigorously? Reviewer #1: Yes[,] Reviewer #2: N/A”

“3. Have the authors made all data underlying the findings in their manuscript fully available? The PLOS Data policy requires authors to make all data underlying the findings described in their manuscript fully available without restriction, with rare exception (please refer to the Data Availability Statement in the manuscript PDF file). The data should be provided as part of the manuscript or its supporting information, or deposited to a public repository. For example, in addition to summary statistics, the data points behind means, medians and variance measures should be available. If there are restrictions on publicly sharing data—e.g. participant privacy or use of data from a third party—those must be specified. Reviewer #1: Yes[,] Reviewer #2: No”

Thank you for the assessment. Before conducting the interviews, an ethics application was submitted to, and approved by the ethical committee of the Technische Universität Berlin. The application addresses the use of collected data: by signing the informed consent, interview participants confirmed that anonymized quotes based on the anonymized transcripts could be used for publication as part of the research project. In compliance with this requirement, we give single statements of the interview participants in the manuscript (Table 2, p.13; Fig 2) and the supporting information (S8, S9 and S10 Files, and S5 Table) for traceability and support of our study findings. We are not allowed to offer access to anonymized interview transcripts in another format, extend, or on demand. According to the informed consent personal data of the participants is only presented in aggregated form so that no conclusions can be drawn about individual persons. For this reason, we do not provide the data set on the results of the survey on socio-demographic characteristics by questionnaire.

“4. Is the manuscript presented in an intelligible fashion and written in standard English? PLOS ONE does not copyedit accepted manuscripts, so the language in submitted articles must be clear, correct, and unambiguous. Any typographical or grammatical errors should be corrected at revision, so please note any specific errors here. Reviewer #1: Yes[,] Reviewer #2: No”

We have proof-read the manuscript again and corrected remaining typographical and grammatical errors.  

Reviewer #1: 

“The authors present a study about reasons of choice in dental care treatment. In Germany, every adult visits the dentist 2.4 times a year on average. In addition to paid preventive care, there are four reasons for this: pain, functional limitations, appearance and psychosocial impairments. Many older people have dentures (that is good). Sometimes over-treatment takes place (that is bad). For some groups in the population, however, there are barriers to using dental care: children, the very old, the disabled, multimorbid (ASA 3+), pain or fear patients etc. For these groups we need qualitative studies in order to break down the barriers but not for healthy middle-aged women to increase the risk of over-treatment. The topic and the methodology of the paper are ok, only the target group is unsuitable in the opinion of the reviewer.”

We fully agree that especially those patient subgroups you mentioned should be part of qualitative studies. However, we think that it is also important to know what criteria are used by patients when deciding for or against a dental treatment. Even if there is full coverage for basic dental care and partly coverage for dentures (basic dentures, defined by the Federal Joint Committee), some treatments in dental care and especially dentures can be very expensive. Additionally, patients have the choice between different materials and treatments. Compared to inpatient care for instance, patients have high out-of-pocket payments. In inpatient care physicians finally decide on the product/procedure. Regarding dentures there are different opportunities for patients (e.g., implant vs. bridge). Physicians may suggest a treatment, but the final decision is taken by patient and may have different reasons: e.g. financial burden, risk of procedures etc. To our knowledge there is no study investigating those reasons. Therefore, the aim of our study was to identify patients’ reasons for choosing or not choosing a dental treatment, following an inductive approach and without a restriction to patients with special needs or diseases. However as mentioned, we fully agree that dental care of patient subgroups (e.g., children, patients with depression disease) should be part of future research. Thank you for this important hint. 

 

Reviewer #2: 

1 Summary of the research

“The main research question of the current manuscript is twofold: (a) What factors influence patients in Germany in their decision on dental treatment? (b) How do patients assess cost coverage of dental services by German statutory health insurance? The authors claim to have identified (a) themes affecting patients’ decision-making on dental treatment and (b) financial challenges of dental treatments for the patients and ways how they take action against them. The authors draw the following conclusions: There are different reasons for choosing or not choosing dental treatment which can be categorised hierarchically into two main categories with several sub-categories each. Furthermore, financial aspects play a major role in decision-making for or against dental treatment and patients employ different measures to alleviate this financial burden. One strength of this study is that the research question is probably of interest for many different audiences, e. g. researches, practitioners, policy makers, and patients alike. Moreover, the approach is suitable for identifying (until today) unknown reasons for (not) choosing dental treatment. Finally it is to be welcomed that the authors used member checking as a quality assurance measure, i. e. they asked the participants after a break to confirm, reject, and complete the identified reasons in order to obtain a “peer-approved” list of reasons. 

But the research questions also bear weaknesses: If these questions would have been formulated in a more concrete manner more valuable insights could have been revealed. Further, the authors made not full use of the strengths of the qualitative methods employed. As a consequence the results remain needlessly superficial: It seems highly likely that new insights can be revealed from the data if additional data analysis steps would be taken. 

On the whole I recommend to accept the manuscript for publication after a major revision.”

Thank you very much for your detailed comment. According to your advice, which is really helpful, we did a major revision and included a more detailed analysis. Unfortunately, we cannot change the research questions. We think that the type of our questions fits well with the aim of the study. 

2 Examples and evidence

2.1 Major issues

“There is one major issue which should be addressed by the authors primarily. Before I begin with the major issue I present two minor issues because all three of them are intertwined. Further minor issues follow in the next section.

I suggest changing the manuscript title from “factors of choice in dental care treatment” to “reasons for (not) choosing dental care treatment”. “Factors of choice” seems misleading since (a) it could be understood as “factors which should be chosen”; (b) it insinuates a quantitative approach instead of a qualitative one: In the manuscript the researchers do not analyse causal factors derived statistically but subjective reasons which people give for their actions.

Thank you very much for the suggestion. We changed the manuscript title to “Reasons for (not) choosing dental treatments – A qualitative study based on patients’ perspective”.

I suggest changing the first research question from “What factors influence patients in Germany in their decision on dental treatment?” to “What reasons do patients in Germany give for choosing or not choosing dental treatment?”. Mentioned themes found in transcripts are interesting per se but the elephant in the room is the question whether these themes represent reasons in favour of or against health care utilisation. I suggest changing the second research question from “How do patients assess cost coverage of dental services by German statutory health insurance?” to “What do patients think about cost coverage of dental services by German statutory health insurance?”. The term “assess” is ambiguous: It could mean “how do they think about cost coverage” (in the sense of “associations/ideas”) or it could mean “how do they evaluate cost coverage” (in the sense of “good/bad”). The reported results imply that the former interpretation seems adequate.

We found your suggestion very helpful and changed the wording of research question no.1 “(1) What reasons do patients have for choosing or not choosing a dental treatment?” but kept the wording of research question no.2 since we think “assess” fits well. The wording in the manuscript regarding the research question no.1 was adjusted accordingly.

One major issue arises from the data analysis. The manuscript text in the results section indicates that the focus group participants did not only name themes which revolve around dental health care utilisation but also whether the mentioned aspects represent reasons for choosing or not choosing dental treatment. It seems to suggest itself that this crucial dichotomy should have been coded and analysed as well: Firstly, I propose to use the additional codes “reason for choosing dental treatment” and “reason for NOT choosing dental treatment”. Secondly, it should be analysed which of the already identified reasons occur mainly in conjunction with the code “reason for choosing dental treatment” and with the code “reason for NOT choosing dental treatment” – and which reasons seem to be ambivalent because some people give these reasons for choosing dental treatment and others for NOT choosing dental treatment. It would be heuristically instructive if the authors would visualise the results like in figure 2 including the categories “group of themes mainly interpreted as reasons for choosing dental treatment” and “group of themes mainly interpreted as reasons for NOT choosing dental treatment” and a third category “group of ambivalent themes, sometimes interpreted in favour of and sometimes against dental treatment”.

This is an important issue: In the realm of quantitative research it would be like having large-scale survey data suited for multivariate inferential statistical analysis and analysing only descriptive frequency distributions and cross-tabulations.”

We strongly support the idea to further categorize the identified findings into “reasons for choosing dental treatment” and “reasons for NOT choosing dental treatment” and therefore continued the data analysis. We rescreened the statements already given in the subcategories for each reason. We used the already existing codes and sorted the participants’ statements according to the reason for choosing and not choosing a dental treatment. If a statement could not be clearly assigned, it was defined as "unclear". In the following step, we added information on the interview questions to the statement. The “unclear” statements were screened again and mostly (re)sorted to “reasons for choosing a dental treatment” and “reasons for NOT choosing a dental treatment”. The screenings were performed independently by the researchers (SF, MB) for the statements of interview group 1 as a pilot. Intercoder agreement of r≥0.8 was considered as acceptable. Consensus was reached and SF continued sorting for interview groups 2 to 4. The number of statements was then counted and analyzed. We added an overview on statements distributions to the supporting information (S7 Table). However, we decided to use a reader-friendly table instead of listing the number of statements to the 53 reasons. According to the aim of qualitative research, we think this is a good way of presenting our additional results. 

The methodological procedure and findings were added to the methods section and the results section and are further discussed in the discussion section of the manuscript. One finding is that many reasons can be classified as ambivalent, i.e., they are used for and against choosing a dental treatment. 

2.2 Minor issues

“In general, “factors of choice in dental treatment” should be replaced by “reasons for (not) choosing dental treatment”. Is the term “performance” the adequate term for the subcategories and codes which this category contains? “Health care service” seems to comprise the meanings better.

The detailed excursus on the coverage of dentures in Germany should be explicitly framed as additional information so readers have more context to understand the results (if that is its purpose). Otherwise this excursus does not seem relevant for the manuscript since neither the data generation process (focus groups) nor the data analysis process nor the results focus on dentures.

The methods should be explained in more detail, to make the way of the data more comprehensible, which steps were taken and so on. The authors should take heed of the “Standards for Reporting Qualitative Research: A Synthesis of Recommendations” by O’Brien et al. (2014). Especially, (a) they should reflect more on the sampling strategy: Why were participants recruited that way and what are the implications of this approach? (b) For the readers the main questions from the manual for the focus groups are necessary to gain a better grasp of the data generation process. (c) Relevant excerpts from the codebook or coding-scheme could be shown to retrace the data analysis process.

Data availability is limited due to participant privacy. This is a common issue with qualitative data but the authors offer access to anonymised interview transcripts on demand.

Finally, further proof-reading is necessary: There are still some orthographical and grammatical errors left.”

Thank you for the helpful comments. We replaced the wording “factors of choice in dental treatment” by “reasons for (not) choosing dental treatment/s” (or “factor”/”factors” by “reason”/”reasons”) throughout the whole manuscript including figures, tables, the supporting information and the corresponding titles. We also replaced the word “performance” by “health care service” in the different sections. Further, we added the wording “Excursus:” to the heading of the paragraph giving additional information on coverage of dentures in Germany. Also, we added a short introduction to this paragraph to clearly outline that this section gives additional information (p.4 in the manuscript).

We now follow the criteria of the COREQ (COnsolidated criteria for REporting Qualitative research) checklist [published in “Consolidated criteria for reporting qualitative research (COREQ): a 32-item checklist for interviews and focus groups” by Tong et al. (2007)]. Accordingly, we partly extended the manuscript, e.g. methods section. We filled out in the COREQ checklist and provide it as supporting information (S1 File). In this respect, we added information about the interviewers/researchers and our data analysis process in more detail. Furthermore, we added information on our sampling strategy. Also, we added the interview guide to the supporting information (S1 Table). We used the recommended article of O’Brien et al. (2014) among other literature to report our research in the best possible way. 

By signing the informed consent, interview participants confirmed that anonymized quotes based on the anonymized transcripts may be used for publication as part of the research project. In compliance with this requirement, we give single statements of the interview participants in the manuscript (Table 2, p.13; Fig 2) and the supporting information (S8, S9 and S10 Files, and S5 Table) for traceability and support of our study findings. We are not allowed to offer access to anonymized interview transcripts in another format, extend, or on demand. With the aim to give another opportunity to reproduce our data analysis process we provide 

• a schematic presentation of the coding-tree, and the coding scheme including our codebook in the supporting information (S2 Table), 

• the descriptive collection and analysis of “most”-important reasons (S7 File)

• the data base for calculating the intercoder-agreements of sorting statements (S8 and S9 File), and

• the data base for analysis of distribution of sorted statements on reasons (S10 File).

We have proof-read the manuscript again and corrected typographical and grammatical errors.

---

## [Decision Letter · Decision Letter 1]

8 Mar 2022

PONE-D-20-35002R1Reasons for (not) choosing dental treatments – A qualitative study based on patients’ perspectivePLOS ONE

Dear Dr. Felgner,

Thank you for submitting your manuscript to PLOS ONE. After careful consideration, we feel that it has merit but does not fully meet PLOS ONE’s publication criteria as it currently stands. Therefore, we invite you to submit a revised version of the manuscript that addresses the points raised during the review process.

Having intensively double checked your re-submitted draft, two of our external reviewers were satisfied with your revisions. However, a third international expert again has forwarded some further recommendations. Moreover, I also have inspected your revision (see R #4), to come to a more balanced decision. Please note that your manuscript still would not seem satisfying, and is not considered ready to proceed. Indeed, some most critical aspects would seem in need of a thorough discussion. With your re-revision, you should follow the reviewers' comments added below, to finalize your paper convincingly, and to meet both Plos One's quality standards and our readership's expectations. 

In order to expedite the processing of the revised manuscript, please make sure to address each of the criticized aspects and incorporate all your carefully elaborated responses within the manuscript. Your rebuttal letter should include point-by-point responses to the reviewer comments, even if you happen to disagree with them, or feel not being able to incorporate all the suggested feedback. Thus, I would like to encourage you to provide a thorough (in terms of language, reviewers' constructive criticism, content, generalizable outcome, and/or Authors' Guidelines) revision in order to avoid an iterative and lengthy review process and facilitate a smooth publication process. Please remember that Plos One will not provide an in-depth copy-editing service, and this requires flawless manuscripts to proceed.

We look forward to receiving your revised manuscript.

Kind regards,

Andrej M. Kielbassa, Prof. Dr. med. dent. Dr. h. c.

Academic Editor

PLOS ONE

Reviewers' comments:

Reviewer's Responses to Questions

**Comments to the Author**

1. If the authors have adequately addressed your comments raised in a previous round of review and you feel that this manuscript is now acceptable for publication, you may indicate that here to bypass the “Comments to the Author” section, enter your conflict of interest statement in the “Confidential to Editor” section, and submit your "Accept" recommendation.

Reviewer #3: All comments have been addressed

Reviewer #4: (No Response)

Reviewer #5: All comments have been addressed

Reviewer #6: All comments have been addressed

2. Is the manuscript technically sound, and do the data support the conclusions?

Reviewer #3: Partly

Reviewer #4: No

Reviewer #5: Yes

Reviewer #6: Yes

3. Has the statistical analysis been performed appropriately and rigorously? 

Reviewer #3: N/A

Reviewer #4: No

Reviewer #5: Yes

Reviewer #6: Yes

4. Have the authors made all data underlying the findings in their manuscript fully available?

Reviewer #3: Yes

Reviewer #4: Yes

Reviewer #5: Yes

Reviewer #6: No

5. Is the manuscript presented in an intelligible fashion and written in standard English?

Reviewer #3: Yes

Reviewer #4: Yes

Reviewer #5: Yes

Reviewer #6: Yes

6. Review Comments to the Author

Reviewer #3: I have read you manuscript with great interest. I do have some comments which hopefully enhance your submission:

- You set up the context of the provision of dental care in Germany well. Although it is quite a long introduction. Could be condensed

- I would remove cross-sectional study as this is a qualitative study and this is likely to confuse the reader.

- Do you have a reference for the ethical approval?

- How many pilot interviews did you do, as more than 1-2 is starting the qual analytical process

- You need to have more detail around the rationale and process of the researchers discussions during the '20min break'

- personally I am not sure you can use an inter-rater reliability in a very subjective methodological approach - a consensus decision between the research team is more relevant for qual studies

- I am not sure of the 'general' analytical approach you have taken - could you make it clearer? It appears you have done a form of content analysis (reporting a quantitative approach to the data generated)

- You don't need an average age, the range is fine

- You need to explain Table 1 is the participants in each Focus Group as in isolation it is a confusing table

- I would not link the 53 reasons and categories, as these are two very different approaches...it is confusing. You're qualitative analysis has found "Two categories, each with 4 sub-categories" and the content analysis, whereby 53 individual reasons were identified.

- I would like, and expect, direct quotations to be included in a results sections for the categories. The results section is quite confusing, and doesn't flow well. Personally, I think you need to focus more on what data is telling you, backing up the categories/subcategories with verbatim quotes.

- Your discussion in general is ok, I think you need to link your findings more with the data/quotes you have obtained. This would enrichen the data.

- You also need to consider reflexivity of the researchers and how this influenced the focus groups/analysis. I would suggest you include this, and reference appropriate. My colleague (Geddis-Regan, 2021) has written a paper in JDR-CTR about this specific issues for clinician-researchers in dentistry and this would support this paper.

Reviewer #4: General comment

- "To our knowledge, no study so far has investigated decision-making of patients in dental care in Germany. Therefore, this study will make a strong scientific contribution by exploring those factors." From the perspective of an international readership, this would neither seem intriguing nor exciting. Remember that Germany's regulations would not seem comparable to other countries.

Abstract

- Word maximum is 300 here, so please add more detailed information to attract the reader. Remember that our future readers will switch to the full text AFTER having read your Abstract section.

- "We conducted four focus group interviews with 27 participants (...)." 27 participants would neither seem convincing nor representative. Thus, when referring to this draft's validity, the content of this paper would seem questionable. Please note that this aspect already has been commented on by the previous Editor. One of the previous reviewers has stated that "the target group is unsuitable", and this would seem right.

- Phrases like "Challenges of financing costly treatments include (...)." or "Most-important reasons determined by patients comprise (...)." do not refer to exact results, but would seem vague only. Please provide precise outcome.

- With your Conclusions, please stick exclusively to your aims. Do not simply repeat your results here. Do not give a summary here. Instead, provide a reasonable and generalizable extension of your outcome, which must be based on your results.

- Remember that such a study would seem interesting for the German reader only, and, thus, might be publishable in a more regional Journal. In case you want to go for an international Journal, you must provide aspects relevant for an international auditorium. Indeed, I do agree with the previous Editor: "Berlin is not representative for Germany."

Intro

- Do not use Authors' names with your text (remember that previous work will be acknowledged with your references). Instead, please focus on your main thoughts.

Meths

- Inviting "Participants (...) via different channels to attract individuals with various patient characteristics [call via Facebook, eBay; a mailing-list of the Technische Universität Berlin (TUB); flyers in grocery stores across all districts of Berlin]." would not seem convincing. Please elaborate more clearly how you have tried to compensate for a non-biased group of respondents.

- Total number of invited participants missing.

Results

- "Overall, 37 people registered for the interviews." The would seem a low number, not considered representative, neither for Berlin, nor for Germany.

- This is confirmed by Table 1 (participants’ characteristics).

- Groups 1 to 4 still would seem unclear.

- As a more general comment, results as given ("preferred fast hardening tooth fillings", "decision-making including out-of-pocket payments and income", "cost-benefit of a treatment", "dental supplementary insurance influencing the decision", and others) would not seem surprising. Please elucidate what would be new here.

- Same with the other aspects given with this section.

Disc

- "This study provides detailed insights to patients’ reasons (...)." Detailed insight to 27 Berlin patients' responses would not seem reliable.

- "Our findings correspond to the results of previous studies (...)." Again, this would seem confirmatory only, right?

- "Additionally, participants reported, they request a second offer on 334 treatments and costs from another dentist." Indeed, this aspect refers to to a few respondents. Do you really think that this is a major aspect to discuss?

- "Due to the study design and the recruitment methods, results may not be representative for the German population." Why do you say "may" here?

Concl

- Again, with your Conclusions, please stick exclusively to your aims. Do not simply repeat your results here. Do not give a summary here. Do not provide any speculations. Instead, provide a reasonable and generalizable extension of your outcome, which must be based on your results.

Refs

- Please revise for uniform formatting. Consulting some recently published Plos One papers would seem helpful.

- Style would be "Ástvaldsdóttir Á, Åhlund K, Holbrook WP, de Verdier B, Tranæus S. Approximate caries detection by DIFOTI: In vitro comparison of diagnostic accuracy/efficacy with film and digital radiography. Int J Dent. 2012; 2012: 326401. https://doi.org/10.1155/2012/326401 PMID: 23213335"

In total, this revised and re-submitted draft still would not seem convincing. Please note that Plos One ask for a sound scientific rationale for the submitted work, and this has not been convincingly elaborated; confirmatory submissions (replicating existing work will likely be rejected if authors do not provide adequate justification). Information as presented is based on a very small number of participants only, and this would not seem reliable, or even representative.

Reviewer #5: (No Response)

Reviewer #6: The present manuscript highlights certain factors that influence patients in Germany in their decision on dental treatment and the way patients assess cost coverage of dental services by German statutory health insurance.

Focus group interviews - that were included in the study design - can be considered valuable tools in identifying issues that are more difficult for the community to become aware of. Such an approach allows the acquisition of "unfiltered" data directly from real life.

The authors reworked and analysed the data presented in the original manuscript and rewrote important parts of it, according to the reviewers’ suggestions.

The authors have described and clearly explained in the manuscript the limitations of this study, (e.g., regarding the selection of participants: “ […] certain behaviors concerning the utilization of dental treatments […] may differ between regions”). The authors have also mentioned that dental care of patient subgroups (e.g., children, patients with depression disease) should be part of future research.

Additionally, the authors replaced the wording “factors of choice in dental treatment” by “reasons for (not) choosing dental treatment/s” (or “factor”/”factors” by “reason”/”reasons”) throughout the whole manuscript including figures, tables, the supporting information and the corresponding titles. All these mentioned modifications are welcome.

Regarding Data Availability - the authors mentioned and explained that some restrictions will apply.

The manuscript is technically sound, and the data support the conclusions. The study design and the included analysis have been performed appropriately, the manuscript is presented in an intelligible fashion and written in standard English.

7. PLOS authors have the option to publish the peer review history of their article (what does this mean?). If published, this will include your full peer review and any attached files.

Reviewer #3: **Yes: **Greig D Taylor

Reviewer #4: No

Reviewer #5: No

Reviewer #6: No

---

## [Author Response · Author response to Decision Letter 1]

8 Apr 2022

Dear Prof. Dr. Kielbassa,

First, we would like to thank you and your team for giving us the opportunity to revise the manuscript. We addressed all of the reviewers' concerns carefully and have responded to each recommendation as directly as possible. We have found your and the reviewers’ comments to be very insightful and helpful and feel that the manuscript has greatly benefited from it.

Below please find our responses (in non-bold letters) to each of the points raised by you and the other reviewers (in bold letters). 

We have included a marked-up copy of our manuscript highlighting the changes to the original version ("Revised Manuscript with Track Changes,") as well as an unmarked version ("Manuscript").

In addition to the revisions, we would like to ask for a name change of one author. Ms Marie Böcker now has the surname Dreger.

Please do not hesitate to contact us if anything remains unclear or further revisions are needed. 

Once again, we thank you very much for your time! 

Yours sincerely,

Susanne Felgner

Editor’s comments

Please see answers to Reviewer #4.

Journal Requirements

None.

Reviewers’ comments to the author

“1. If the authors have adequately addressed your comments raised in a previous round of review and you feel that this manuscript is now acceptable for publication, you may indicate that here to bypass the #Comments to the Author” section, enter your conflict of interest statement in the “Confidential to Editor” section, and submit your "Accept" recommendation.”; Reviewer #3, #5 & #6: “All comments have been addressed”, and “Reviewer #4: (No Response)”

Thank you very much for the assessment.

“2. Is the manuscript technically sound, and do the data support the conclusions? The manuscript must describe a technically sound piece of scientific research with data that supports the conclusions. Experiments must have been conducted rigorously, with appropriate controls, replication, and sample sizes. The conclusions must be drawn appropriately based on the data presented. Reviewer #3: Partly [,] Reviewer #4: No”; Reviewer #5 & #6: “Yes”

Thank you for the assessment. To meet the stated requirements, we have revised the manuscript regarding the following points: we have revised the number of responses in the recruitment process and the number of participants in the methods section, since we now include the pilot interview in this regard. To ensure the rigorous implementation of our study, we applied the Consolidated Criteria for Reporting Qualitative Research (COREQ) checklist. Details are given in the methods section. The documentation for COREQ can be found in the supporting information (S1 File). In accordance with the comments of reviewer #3, we give verbal quotes of the study’s participants in the results section. In the discussion section we particularly emphasize limitations of qualitative studies. Nevertheless, we underline the explorative character of this approach. In the conclusion section we refer to our findings. However, we do not repeat results, except when giving examples. In the results section, the discussion section and the conclusion section we do not use general statements or draw general conclusions. To make the manuscript interesting for the international readership, we have revised the abstract. “Oral health is increasingly seen as a public health challenge due to the remarkable prevalence of oral diseases worldwide, the impact on general health, and health consequences that can arise for individuals. Compared to other health services, oral health services are usually not fully covered by statutory health insurance, which is seen as one reason in decision-making on dental treatments. Nevertheless, patients’ reasons for treatment decisions are not well understood although they can provide valuable insights.” (p. 2, line 23 ff.). Also, we have revised the introduction. “Previous studies considered costs, long duration times and patients’ fear as reasons of choice for dental treatments in different countries, e.g., Saudi-Arabia [12], and the USA [13]. Some existing studies are limited to certain dental services, e.g., preventive measures [14], and caries treatment [15]. However, a basic understanding of possible reasons of choice for dental treatments from patients’ perspective is limited, especially in Germany. Although this study focuses on the German health care setting, it contributes to the identification of reasons that may influence a decision for dental treatments.” (p. 3, line 67 ff.).

“3. Has the statistical analysis been performed appropriately and rigorously? Reviewer #3: N/A [,] Reviewer #4: No”; Reviewer #5 & #6: “Yes”

Thank you for the assessment. All parts of the descriptive analysis were checked and revised according to the reviewers’ comments. In this context, Table 1 was revised. Although this point does not fully apply to qualitative studies, we have highlighted particular points with regard to the quality of qualitative studies. Aiming at an appropriate and a rigorous implementation of our study, we followed the Consolidated Criteria for Reporting Qualitative Research (COREQ) checklist. The documentation for COREQ can be found in the supporting information (S1 File). Furthermore, limitations of our study are mentioned and discussed in the manuscript. “The findings of this study must be considered in the light of limitations linked to the methods, and to the recruiting process that is restricted to specific channels and one region. Although, focus group interviews have the opportunity of thought-provoking impulses within a discussion, this can also lead to a restrained behavior of participants. They may not express their overall opinions. Due to our recruitment process selection bias might have been occurred [63]. In addition, because of the small number of participants, which is in the nature of the method, the sample of our participants is not representative for the German population [64]. Furthermore, our approach of purposive sampling excludes the claim of representativity; however, it enables us to gain "information-rich" cases with experience and interest in the topic [65]. We indicate that certain behaviors concerning the utilization of dental treatments (e.g., oral health behavior, decision-making, or willingness-to-pay) may differ between regions. Similarly, participants with a university degree were above the German average while unemployed participants were slightly below [66]. Most participants in many dental qualitative studies are female, like in our study [67]. Also, reflexivity of the researchers should be briefly assessed. It can be confirmed that all researchers have already had experience as patients in dental care. Due to research and teaching activities, SF and CH also had knowledge of the German healthcare system including dental care. This, and experiences in the course of the study, could have had an influence on implementing and analyzing the interviews [68, 69].” (p. 19 f., line 455 ff.).

“4. Have the authors made all data underlying the findings in their manuscript fully available? The PLOS Data policy requires authors to make all data underlying the findings described in their manuscript fully available without restriction, with rare exception (please refer to the Data Availability Statement in the manuscript PDF file). The data should be provided as part of the manuscript or its supporting information, or deposited to a public repository. For example, in addition to summary statistics, the data points behind means, medians and variance measures should be available. If there are restrictions on publicly sharing data—e.g. participant privacy or use of data from a third party—those must be specified.”; Reviewer #3, #4, & #5: “Yes”; “Reviewer #6: No”

Thank you for the assessment. We would like to kindly point out that due to data protection issues we do not publish the transcripts of the focus group interviews. Information on this has been given in the submission form: “All relevant data that can be publicly displayed are available within the paper and its Supporting Information files. We do not offer access to anonymized interview transcripts in another format, extend, or on demand. Non-aggregated participants information and transcripts may not be publicly shared due to ethical restrictions, approved by the ethical committee of the Technische Universität Berlin in Berlin, Germany.”.

“5. Is the manuscript presented in an intelligible fashion and written in standard English? PLOS ONE does not copyedit accepted manuscripts, so the language in submitted articles must be clear, correct, and unambiguous. Any typographical or grammatical errors should be corrected at revision, so please note any specific errors here.”; Reviewer #3, #4, #5, & #6: “Yes”

Thank you for the assessment.

Reviewer #3

“I have read you manuscript with great interest. I do have some comments which hopefully enhance your submission: - You set up the context of the provision of dental care in Germany well. Although it is quite a long introduction. Could be condensed - I would remove cross-sectional study as this is a qualitative study and this is likely to confuse the reader. - Do you have a reference for the ethical approval? - How many pilot interviews did you do, as more than 1-2 is starting the qual analytical process - You need to have more detail around the rationale and process of the researchers discussions during the '20min break' - personally I am not sure you can use an inter-rater reliability in a very subjective methodological approach - a consensus decision between the research team is more relevant for qual studies - I am not sure of the 'general' analytical approach you have taken - could you make it clearer? It appears you have done a form of content analysis (reporting a quantitative approach to the data generated)”

Thank you for the assessment. The introduction was shortened and strengthened. In the methods section the wording “cross-sectional study” was replaced by “qualitative study” (p. 5, line 119). Furthermore, the reference number for the ethical approval was added (p. 6, line 132).

One pilot interview was conducted. We did not consider the pilot in our qualitative analysis. We added respective information regarding the 20-minute break. “During a 20-minute break, the discussion protocol of each focus group interview was screened by the two researchers to summarize statements belonging to the same content to identify reasons for (not) choosing dental treatments that were inductively derived from it.” (p. 6 f., line 154 ff.).

Regarding intercoder-reliability (ICR) we would like to maintain the results of the analysis. ICR is a tool to precisely ensure the trustworthiness of reviewers involved in coding, because the methodological approach is subjective. In qualitative research, including focus group interviews, ICR is recommended by various sources [1], and implemented in latest research [2, 3]. We agree that a consensus discussion is an important approach within an analyzing process. We therefore also used joint discussions to reach consensus. To clarify, we revised all relevant sentences: “If necessary, codes were refined using joint discussions of the two researchers.” (p. 8, line 188 f.), “In a further qualitative analysis, (sub)categories on the reasons of choice were formed based on joint discussion between the researchers.” (p. 8, line 190 f.), and “After consensus decision was reached in a subsequent discussion, SF continued screening and sorting for the remaining interview groups.” (p. 8, line 199 ff.).

Regarding our analytical approach, we used conventional content analysis as one approach of qualitative content analysis. Using this inductive approach, codes and (sub)categories emerge from the data, and findings move from specific to general. We have revised the corresponding sentence so that this information is clearly addressed. Additionally, we cite basic literature regarding this method [4]. “Through this inductive approach of content analysis, codes of reasons of choice and (sub)categories emerged from the data, and findings moved from specific to general [30, 31]. Content analysis further gives the opportunity to quantify data of qualitative analyses [32].” (p. 7 f., line 178 ff.).

“- You don't need an average age, the range is fine - You need to explain Table 1 is the participants in each Focus Group as in isolation it is a confusing table”

Thank you for the assessment. We have removed the information on average age from the text. In Table 1, we have swapped the information on age range and average age for consistency. In addition, we added a text part to introduce Table 1 that roughly describes its contents. “Descriptive information on the participants per focus group and in total are shown in Table 1.” (p. 9, line 218). We additionally added subheadings within the table (i.e., "Socio-demographic characteristics of participants (n=27)", "Use of incentive measures and supplementary insurance in dental care by participants (n=26)", and "Experiences of participants with out-of-pocket payments (n=27) " (p. 9 f.) and strengthened the table title “Descriptive information on participants per focus group and in total.” (p. 9, line 219). We also added the legend of the table in a bottom row. 

“- I would not link the 53 reasons and categories, as these are two very different approaches...it is confusing. You're qualitative analysis has found "Two categories, each with 4 sub-categories" and the content analysis, whereby 53 individual reasons were identified. - I would like, and expect, direct quotations to be included in a results sections for the categories. The results section is quite confusing, and doesn't flow well. Personally, I think you need to focus more on what data is telling you, backing up the categories/subcategories with verbatim quotes.”

Thank you for this very helpful advice. We now clearly describe the methodological approach in the methods section (p. 7 f., line 177 ff.). Furthermore, we provide introductory words in the results section to the subsequent presentation of combined results in that chapter. “The first category includes reasons of choice for dental treatments focusing on the service of dental care and its characteristics. It is further subdivided into subcategories “preconditions”, “treatment”, “costs”, and “outcomes”.” (p. 10, line 232 ff.), and “The second category includes reasons of choice for dental treatments that focus on the professionals performing dental treatments as well as office structures and processes. Subcategories include "professional skills" and "social skills" of dentists as well as "office staff and equipment", and "office processes”.” (p. 12, line 274 ff.). As you suggested, we added direct quotations (from Figure 2) to the categories and subcategories mentioned. We additionally revised Figure 2 by removing presented quotations to avoid duplicates. The text of the results section has been reworded and the subsections have been structured systematically according to the two categories and four subcategories to improve the reading flow (p. 10 ff., line 231 ff.). The analysis of “’Most-important’ selected reasons of choice” has been revised in the methods section (p. 7, line 160 f.) and removed from the results section to reduce word number, and because we think that the findings have a more informative than results-presenting character. Nevertheless, the results with participants’ “most important” reasons of choice are comprehensible for the reader as we added the corresponding figure to the supporting information (S6 File). In the results section we give the information that the results and figures for the analysis of the “most important” reasons mentioned can be found in the supporting information (S6-8 Files). “Additionally, a descriptive presentation of the number of reasons ranked as "most important" by the participants, a photograph of the chart board from group 3, and a descriptive collection and analysis of reasons can be found in the supporting information (S6, S7, and S8 Files).” (p. 10, line 228 ff.). 

“- Your discussion in general is ok, I think you need to link your findings more with the data/quotes you have obtained. This would enrichen the data. - You also need to consider reflexivity of the researchers and how this influenced the focus groups/analysis. I would suggest you include this, and reference appropriate. My colleague (Geddis-Regan, 2021) has written a paper in JDR-CTR about this specific issues for clinician-researchers in dentistry and this would support this paper.”

Thank you for the helpful comment and the reference. We have added excerpts from the participants’ direct quotes to the discussion section (p. 15 ff.). We have integrated this quotes in such a way that the reading flow has been further improved. Additionally, we assessed the reviewers' reflexivity within the limitations. “Also, reflexivity of the researchers should be briefly assessed. It can be confirmed that all researchers have already had experience as patients in dental care. Due to research and teaching activities, SF and CH also had knowledge of the German healthcare system including dental care. This, and experiences in the course of the study, could have had an influence on implementing and analyzing the interviews [67, 68].” (p. 20, line 468 ff.). Thanks again for providing the reference, that we considered (p. 20, line 472).

References

1. O’Connor C, Joffe H. Intercoder reliability in qualitative research: debates and practical guidelines. Int J Qual Methods. 2020;19:160940691989922 https://doi.org/10.1177/1609406919899220.

2. Henage CB, Ferreri SP, Schlusser C, Hughes TD, Armistead LT, Kelley CJ, et al. Transitioning focus group research to a videoconferencing environment: a descriptive analysis of interactivity. Pharmacy. 2021;9(3):1-9 https://doi.org/10.3390/pharmacy9030117 PMID: 34202707.

3. Carroll AL, Chan A, Steinberg JR, Bryant TS, Marin-Nevarez P, Anderson TN, et al. Medical student values inform career plans in service & surgery - A qualitative focus group analysis. J Surg Res. 2020;256:636–44 https://doi.org/10.1016/j.jss.2020.07.030 PMID: 32810664.

4. Hsieh H-F, Shannon SE. Three approaches to qualitative content analysis. Qual Health Res. 2005;15(9):1277–88 https://doi.org/10.1177/1049732305276687 PMID: 16204405.

Reviewer #4

“General comment - "To our knowledge, no study so far has investigated decision-making of patients in dental care in Germany. Therefore, this study will make a strong scientific contribution by exploring those factors." From the perspective of an international readership, this would neither seem intriguing nor exciting. Remember that Germany's regulations would not seem comparable to other countries.”

Thank you for your comment. We agree that Germany’s regulation is not comparable to other countries. Nevertheless, it might also be of interest for an international readership, especially in the context of a system comparison. Attention to dental care is increasing (e.g., The Lancet oral health series [1]). Nevertheless, due to its high proportion of out-of-pocket payments dentistry is hardly comparable with other areas in health care. Therefore, it is of great interest to take a closer look at reasons of choice for dental treatments from patients' view and to investigate those in an explorative qualitative way opening up space for further (quantitative) research. Studies are scarce and health expenditures as share of GDP spendings in Germany are increasing although having one of the highest spendings in dental care in the EU [2]. To make the manuscript interesting for the international readership, we have revised the abstract: “Oral health is increasingly seen as a public health challenge due to the remarkable prevalence of oral diseases worldwide, the impact on general health, and health consequences that can arise for individuals. Compared to other health services, oral health services are usually not fully covered by statutory health insurance, which is seen as one reason in decision-making on dental treatments. Nevertheless, patients’ reasons for treatment decisions are not well understood although they can provide valuable insights.” (p. 2, line 23 ff.); and the introduction: “Previous studies considered costs, long duration times and patients’ fear as reasons of choice for dental treatments in different countries, e.g., Saudi-Arabia [12], and the USA [13]. Some existing studies are limited to certain dental services, e.g., preventive measures [14], and caries treatment [15]. However, a basic understanding of possible reasons of choice for dental treatments from patients’ perspective is limited, especially in Germany. Although this study focuses on the German health care setting, it contributes to the identification of reasons that may influence a decision for dental treatments.” (p. 3, line 67 ff.). Accordingly, the discussion section (p. 15 ff.) and the conclusion section (p. 20 ff.) in the text were also revised. Moreover, our manuscript vividly presents the focus group method as a qualitative approach. Therefore, we consider our study to be exciting for many readers.

“Abstract - Word maximum is 300 here, so please add more detailed information to attract the reader. Remember that our future readers will switch to the full text AFTER having read your Abstract section. - "We conducted four focus group interviews with 27 participants (...)." 27 participants would neither seem convincing nor representative. Thus, when referring to this draft's validity, the content of this paper would seem questionable. Please note that this aspect already has been commented on by the previous Editor. One of the previous reviewers has stated that "the target group is unsuitable", and this would seem right. - Phrases like "Challenges of financing costly treatments include (...)." or "Most-important reasons determined by patients comprise (...)." do not refer to exact results, but would seem vague only. Please provide precise outcome. - With your Conclusions, please stick exclusively to your aims. Do not simply repeat your results here. Do not give a summary here. Instead, provide a reasonable and generalizable extension of your outcome, which must be based on your results. - Remember that such a study would seem interesting for the German reader only, and, thus, might be publishable in a more regional Journal. In case you want to go for an international Journal, you must provide aspects relevant for an international auditorium. Indeed, I do agree with the previous Editor: "Berlin is not representative for Germany."”

We rewrote our abstract to also attract an international auditorium (p. 2; please see our response regarding an international readership to your comment above). Furthermore, we now explicitly focus on the participants of the study when describing the results and avoid generalizations (p. 8 ff.); e.g., “Additionally, the participants look for interdisciplinarity in the dental office, in order to have a broad basis of information. They reported to expect a professional treatment and costs information […].” (p. 12, line 284 ff.). 

Furthermore, we clearly point out the challenges of a qualitative study, that are inherent to the methodology, in the limitations. “The findings of this study must be considered in the light of limitations linked to the methods, and to the recruiting process that is restricted to specific channels and one region. Although, focus group interviews have the opportunity of thought-provoking impulses within a discussion, this can also lead to a restrained behavior of participants. They may not express their overall opinions. Due to our recruitment process selection bias might have been occurred [63]. In addition, because of the small number of participants, which is in the nature of the method, the sample of our participants is not representative for the German population [64]. Furthermore, our approach of purposive sampling excludes the claim of representativity […]” (p. 19, line 455 ff.). Usually, qualitative studies are not representative and instead are often used in an explorative way to build a new basis for further studies [3]. However, we also clearly point advantages and opportunities of qualitative studies in general and our study in particular. “[…] however, it enables us to gain "information-rich" cases with experience and interest in the topic [65].” (p. 19, line 462 f.). 

Also, we rewrote our conclusions in the abstract, where we avoid simply repeating our results and now focus exclusively on our aims. We also provide aspects relevant for an international readership. “Identified reasons for choosing dental treatments provide a basis for further studies to quantify the relevance of these reasons from patients' perspective. Based on this, the various reasons identified can be considered in future policies to improve patients’ utilization behavior, which can range from improved information sources to increased incentive measures.” (p. 2, line 45 f.; please also see our response regarding an international readership to your comment above).

“Intro - Do not use Authors' names with your text (remember that previous work will be acknowledged with your references). Instead, please focus on your main thoughts.”

Thank you for pointing this out. The author names have been removed from the introduction, and also the discussion section. Furthermore, the introduction section was shortened so that the focus on our main thoughts is strengthened (p. 3 ff.).

“Meths - Inviting "Participants (...) via different channels to attract individuals with various patient characteristics [call via Facebook, eBay; a mailing-list of the Technische Universität Berlin (TUB); flyers in grocery stores across all districts of Berlin]." would not seem convincing. Please elaborate more clearly how you have tried to compensate for a non-biased group of respondents. - Total number of invited participants missing.”

Thank you for the assessment. We have revised the section on recruitment. “Using different communication channels, participants were recruited through online media [Facebook, eBay, mailing list of the Technische Universität Berlin (TUB)], and print media (in all districts of Berlin: flyers in grocery stores, free local weekly newspaper) to reach a diverse group of respondents (i.e., all ages, educational, and income statuses).” (p. 5, line 123 ff.). We hope it is now clear that we have targeted both online and print media to reach potential participants with different (patient) characteristics (i.e., all ages, educational, and income statuses). Different communication channels were used to minimize chances for bias. Nevertheless, there is a possibility for bias. Restrictions due to bias are addressed in the limitations section. We now clearly mention the possibility of selection bias. “The findings of this study must be considered in the light of limitations linked to the methods, and to the recruiting process that is restricted to specific channels and one region. […] Due to our recruitment process selection bias might have been occurred [63].” (p.19, line 455 ff.). Other approaches, e.g., recruitment in dental offices, might have resulted in other, and maybe more stronger biases, e.g., focus on one dentist, location, or participants’ tendency to accept treatments. We therefore remain convinced of our recruitment approach.

The number of people, that responded on our recruitment activities, is given in the results section. We have added the information, that all registered people were also invited to close this information gap. “Overall, 48 people registered and were invited for the interviews, ten participated in the pilot interview, and 27 participated in the focus group interviews 1-4.” (p. 9, line 207 f.).

“Results - "Overall, 37 people registered for the interviews." The would seem a low number, not considered representative, neither for Berlin, nor for Germany. - This is confirmed by Table 1 (participants’ characteristics). - Groups 1 to 4 still would seem unclear. - As a more general comment, results as given ("preferred fast hardening tooth fillings", "decision-making including out-of-pocket payments and income", "cost-benefit of a treatment", "dental supplementary insurance influencing the decision", and others) would not seem surprising. Please elucidate what would be new here. - Same with the other aspects given with this section.”

Thank you for the assessment. With the planned number of four focus groups and a maximum of ten participants, we were within the recommended range for this qualitative approach [4]. We added this information in the methods section: “Designed as a qualitative study, we conducted focus group interviews planned with a maximum of ten participants each, as recommended in scientific literature [26].” (p. 5, line 119 f.), and the results section: “We conducted four focus group interviews with 27 participants (5-9 participants each, which is in line with recommendations for focus groups [26] that lasted 56-134 minutes.” (p. 9, line 209 ff.). Due to the small number of participants, it is in the nature of the method that focus groups are not statistically representative and may not be representative for a population [5]. However, the method has other advantages, e.g., explore and clarify participants view, and group dynamic [5]. This information is also given in the conclusion section in the text. “[…] focus group interviews have the opportunity of thought-provoking impulses within a discussion […]” (p. 19, line 457). Further focus group interviews would have been planned, if we had not reached the saturation point within analysis of conducted interviews. This is also mentioned in the manuscript. “As the point of data saturation was reached in interview group 2, we did not schedule more than the four planned appointments [37].” (p. 9, line 211 ff.). 

After suggestion by your comment, we have decided to include the number of all registered potential participants in the manuscript. This means that the number of participants of the pilot interview is added. Accordingly, we have made a revision in the text (p. 9, line 207 f.; please see the manuscript excerpt in our response regarding the number of participants on your comment above). 

We interpret your comment "Groups 1 to 4 still would seem unclear." as it is not clear why the groups were not fully staffed. The answer is that participants canceled at very short notice (in some cases on the day of the interview) or did not show up at all and subsequently apologized. Re-staffing was therefore not possible. However, even with the current number of five to nine participants per focus group, we were still within the recommended range in terms of number of participants [4]. To clarify Table 1, we added subheadings within the table (e.g., "Socio-demographic characteristics of participants (n=27)" (p. 9) and strengthened the table title “Descriptive information on participants per focus group and in total.” (p. 9, line 219). Furthermore, we added a sentence to introduce Table 1 and roughly describe its contents. “Descriptive information on the participants per focus group and in total are shown in Table 1.” (p. 9, line 218). We also added the legend of the table in a bottom row.

Many results we have generated may already exist partly in the international literature for other countries. However, due to different study designs and different research questions not all reasons can be found in literature. Additionally, our study showed the reasons in detail and to our knowledge, these results do not exist for Germany. We would also like to point out that Germany is, in an EU-comparison, the country with the most extensive coverage from health insurances and one of the countries with highest expenditures in dentistry [2]. We therefore consider it particularly remarkable that reasons such as "out-of-pocket payment", or "actuals costs" seem to be decisive for patients when deciding against dental treatments. Furthermore, reasons of choice from patients’ perspective in dental care were not obtained in the context of qualitative studies so far, to our knowledge. Accordingly, we consider our study to be necessary and our findings to be new. We have added this information in the introduction: “[…] identifying factors that determine decision-making on dental treatments can help to understand patients’ perspective and their decisions regarding their choice of dental treatments. It can provide a basis for further analysis that may reveal a need for changes in policies and practice.” (p. 3, line 63 ff.), and the discussion section in the text: “While there have been studies focusing on patients’ decision-making with regard to certain dental treatments or subgroups, the key strength of this study is that we captured all adult patient groups. The diversity is strengthened by the fact that the study focused on identifying and understanding patients’ reasons of choice for or against dental treatments in general. In addition to the previously known reasons for choosing dental treatments such as out-of-pocket payments and trust in the dentist, new reasons emerged, e.g., holistic treatment.” (p. 16, line 365). What is also new in our study, is the variety and a high number of identified reasons. We point this out in the conclusion section in the text. “Our participants reported to have various reasons for choosing or not choosing dental treatments (e.g., current complaints, cost-benefit).” (p. 20, line 474 f.).

To make the results section clearer, we have divided it into subcategories within the categories and added headings for each. Furthermore, the various reasons for choosing or not choosing dental treatments provide a basis for further studies to quantify the relevance of these reasons from patients' perspective (e.g., in a willingness-to-pay analysis). We give this recommendation in the conclusions in the abstract: “Identified reasons for choosing dental treatments provide a basis for further studies to quantify the relevance of these reasons from patients' perspective.” (p. 2, line 45 f.); and in the text: “[…] the influence of out-of-pocket payments should be investigated quantitatively to determine patients’ willingness-to-pay for dental treatments.” (p. 17, line 405 ff.), and “As the results must be interpreted in the light of the qualitative study design, further studies should quantify the relevance of the reasons for choosing dental treatments from patients' perspective (e.g., in a willingness-to-pay analysis).” (p. 20, line 488 ff.).

“Disc - "This study provides detailed insights to patients’ reasons (...)." Detailed insight to 27 Berlin patients' responses would not seem reliable. - "Our findings correspond to the results of previous studies (...)." Again, this would seem confirmatory only, right? - "Additionally, participants reported, they request a second offer on 334 treatments and costs from another dentist." Indeed, this aspect refers to to a few respondents. Do you really think that this is a major aspect to discuss? - "Due to the study design and the recruitment methods, results may not be representative for the German population." Why do you say "may" here?”

Thank you for the assessment. We have changed the wording to "our participants" in the discussion section (p. 16, line 363). The sentence’s content is thus correct in any case. Overall, we have revised the manuscript to make it clear that the reported results relate only to participants from our focus groups. 

Furthermore, we now point out that our findings are confirmed by studies from other countries. “Our findings correspond to the results of previous studies, undertaken in other countries, presenting reasons such as language barriers [38], complexity of treatments [39] and time of treatment [12, 13].” (p. 16, line 365). Literature given here is still studies conducted in countries other than Germany. 

To our knowledge, as this can also be found in scientific literature, all relevant quotations are included in results of a qualitative analysis, no matter how often given [6]. Accordingly, less frequently given statements have the same weight as (very) frequently given statements. This approach is supported by the limitation that qualitative studies can be assumed to be not representative. This limitation is due to the nature of qualitative analyses [5]. We use the wording "may not be representative" since basic scientific literature is using this subjunctive (II) [5].

“Concl - Again, with your Conclusions, please stick exclusively to your aims. Do not simply repeat your results here. Do not give a summary here. Do not provide any speculations. Instead, provide a reasonable and generalizable extension of your outcome, which must be based on your results.”

Thank you for the assessment. We have completely revised the conclusion section in the text (p. 20, line 473 ff.). Here you can now find answers to the two research questions. Only a few examples of the results are still given. We have drawn conclusions from our findings and formulated recommendations for it. E.g., “To ensure communication between patient and dentist, dentists should be trained in special programs fostering shared decision-making.” (p. 20, line 484 ff.).

“Refs - Please revise for uniform formatting. Consulting some recently published Plos One papers would seem helpful. - Style would be "Ástvaldsdóttir Á, Åhlund K, Holbrook WP, de Verdier B, Tranæus S. Approximate caries detection by DIFOTI: In vitro comparison of diagnostic accuracy/efficacy with film and digital radiography. Int J Dent. 2012; 2012: 326401. https://doi.org/10.1155/2012/326401 PMID: 23213335"

Thank you for the advice. We have made the adjustments to the literature citations. These are now consistent in the manuscript’s reference list (p. 21 ff.).

“In total, this revised and re-submitted draft still would not seem convincing. Please note that Plos One ask for a sound scientific rationale for the submitted work, and this has not been convincingly elaborated; confirmatory submissions (replicating existing work will likely be rejected if authors do not provide adequate justification). Information as presented is based on a very small number of participants only, and this would not seem reliable, or even representative.”

Thank you for the assessment. We have revised the introduction in abstract and text with the aim that the conduct of the study is scientific rationale. We have done this by pointing out the importance of oral health and stating the prevalence of oral diseases worldwide. Thus, we conclude that reasons of choice for dental treatments need to be investigated to understand patients’ behavior and improve their oral health. “Oral health is increasingly seen as a public health challenge due to the remarkable prevalence of oral diseases worldwide, the impact on general health, and health consequences that can arise for individuals. Compared to other health services, oral health services are usually not fully covered by statutory health insurance, which is seen as one reason in decision-making on dental treatments. Nevertheless, patients’ reasons for treatment decisions are not well understood although they can provide valuable insights. The objective of this study was to identify reasons of choice for dental treatments and to explore patients’ view on cost coverage in Germany.” (p. 2, line 23 ff.). We hope that our manuscript is convincing you after this revision.

Furthermore, we would like to point out that a small number of participants is within the recommended range according to scientific literature [4]. We have given this information in the manuscript (p. 5, line 119 f.; and p. 9, line 209 ff.; please see the manuscript excerpt in our response regarding the number of participants on your comment above). In the literature, it is also written that results from focus group interviews are generally to be assessed as “may not be representative” due to this limitation [4]. Regarding the reliability of our results, we would like to kindly point out that we followed recommendations in scientific literature when conducting the qualitative study [7]. In addition, we have documented the procedure of conducting and analyzing the interviews, following the Consolidated Criteria for Reporting Qualitative Research (COREQ) checklist (please see the supporting information for the COREQ checklist document, S1 File) [8]. We would therefore consider the reliability of our results to be given.

References

1. Watt RG, Daly B, Allison P, Macpherson LMD, Venturelli R, Listl S, et al. The Lancet oral health series: implications for oral and dental research. J Dent Res. 2020;99(1):8–10 https://doi.org/10.1177/0022034519889050 PMID: 31859586.

2. OECD/European Union. Health at a glance: Europe 2020: state of health in the EU cycle. Paris: OECD Publishing; 2020.

3. Guest G, Namey EE, Mitchell ML. Collecting qualitative data: a field manual for applied research. 1 Oliver's Yard, 55 City Road London EC1Y 1SP: SAGE Publications, Ltd; 2013.

4. Krueger RA, Casey MA. Focus groups: a practical guide for applied research. 5th ed. Thousand Oaks, California: Sage; 2014.

5. Kitzinger J. Qualitative research. Introducing focus groups. BMJ. 1995;311(7000):299–302 https://doi.org/10.1136/bmj.311.7000.299 PMID: 7633241.

6. Kuckartz U. Qualitative text analysis: a systematic approach. In: Kaiser G, Presmeg N, editors. Compendium for Early Career Researchers in Mathematics Education. Cham: Springer International Publishing; 2019. p. 181–197. 10.1007/978-3-030-15636-7_8.

7. Hsieh H-F, Shannon SE. Three approaches to qualitative content analysis. Qual Health Res. 2005;15(9):1277–88 https://doi.org/10.1177/1049732305276687 PMID: 16204405.

8. Curtin S, Trace A. Focus group methods in dental research. J Ir Dent Assoc. 2016;62(5):271–5 PMID: 29797837.

Reviewer #5

(No Response)

Reviewer #6

“The present manuscript highlights certain factors that influence patients in Germany in their decision on dental treatment and the way patients assess cost coverage of dental services by German statutory health insurance. Focus group interviews - that were included in the study design - can be considered valuable tools in identifying issues that are more difficult for the community to become aware of. Such an approach allows the acquisition of "unfiltered" data directly from real life. The authors reworked and analysed the data presented in the original manuscript and rewrote important parts of it, according to the reviewers’ suggestions. The authors have described and clearly explained in the manuscript the limitations of this study, (e.g., regarding the selection of participants: “ […] certain behaviors concerning the utilization of dental treatments […] may differ between regions”). The authors have also mentioned that dental care of patient subgroups (e.g., children, patients with depression disease) should be part of future research. Additionally, the authors replaced the wording “factors of choice in dental treatment” by “reasons for (not) choosing dental treatment/s” (or “factor”/”factors” by “reason”/”reasons”) throughout the whole manuscript including figures, tables, the supporting information and the corresponding titles. All these mentioned modifications are welcome. Regarding Data Availability - the authors mentioned and explained that some restrictions will apply. The manuscript is technically sound, and the data support the conclusions. The study design and the included analysis have been performed appropriately, the manuscript is presented in an intelligible fashion and written in standard English.”

Thank you very much for your positive assessment of our revised and resubmitted manuscript!

---

## [Decision Letter · Decision Letter 2]

13 Apr 2022

Reasons for (not) choosing dental treatments – A qualitative study based on patients’ perspective

PONE-D-20-35002R2

Dear Dr. Felgner,

We’re pleased to inform you that your manuscript has been judged scientifically suitable for publication and will be formally accepted for publication once it meets all outstanding technical requirements. Congratulations!

Kind regards,

Andrej M Kielbassa, Prof. Dr. med. dent. Dr. h. c.

Kind regards,

Andrej M Kielbassa

Academic Editor

PLOS ONE

Additional Editor Comments (optional):

Reviewers' comments:

Reviewer's Responses to Questions

**Comments to the Author**

1. If the authors have adequately addressed your comments raised in a previous round of review and you feel that this manuscript is now acceptable for publication, you may indicate that here to bypass the “Comments to the Author” section, enter your conflict of interest statement in the “Confidential to Editor” section, and submit your "Accept" recommendation.

Reviewer #3: All comments have been addressed

Reviewer #4: All comments have been addressed

2. Is the manuscript technically sound, and do the data support the conclusions?

Reviewer #3: Yes

Reviewer #4: Yes

3. Has the statistical analysis been performed appropriately and rigorously? 

Reviewer #3: Yes

Reviewer #4: Yes

4. Have the authors made all data underlying the findings in their manuscript fully available?

Reviewer #3: Yes

Reviewer #4: Yes

5. Is the manuscript presented in an intelligible fashion and written in standard English?

Reviewer #3: Yes

Reviewer #4: Yes

6. Review Comments to the Author

Reviewer #3: Thank you for addressing my comments. I still don't think an inter-rater score should be included, however, I appreciate your comments on why you want to keep this in.

Reviewer #4: This revised and re-submitted draft would seem satisfying now, even if there still are some doubts because of the low number of participants.

7. PLOS authors have the option to publish the peer review history of their article (what does this mean?). If published, this will include your full peer review and any attached files.

Reviewer #3: **Yes: **Greig D Taylor

Reviewer #4: No

---

## [Editor Report · Acceptance letter]

17 May 2022

PONE-D-20-35002R2 

Reasons for (not) choosing dental treatments – A qualitative study based on patients’ perspective 

Dear Dr. Felgner:

I'm pleased to inform you that your manuscript has been deemed suitable for publication in PLOS ONE. Congratulations! Your manuscript is now with our production department. 

Kind regards, 

on behalf of

Prof. Dr. med. dent. Dr. h. c. Andrej M Kielbassa 

Academic Editor

PLOS ONE